# DREAMCRAFT3D: HIERARCHICAL 3D GENERATION WITH BOOTSTRAPPED DIFFUSION PRIOR

**Jingxiang Sun**[1][*]**, Bo Zhang**[3][†]**, Ruizhi Shao**[1]**, Lizhen Wang**[1]**, Wen Liu**[2]**, Zhenda Xie**[2]**, Yebin Liu**[1][†]
[1] Tsinghua University, [2] DeepSeek AI, [3] Zhejiang University

## ABSTRACT

We present DreamCraft3D, a hierarchical 3D content generation method that produces high-fidelity and coherent 3D objects. We tackle the problem by leveraging a 2D reference image to guide the stages of geometry sculpting and texture boosting. A central focus of this work is to address the consistency issue that existing works encounter. To sculpt geometries that render coherently, we perform score distillation sampling via a view-dependent diffusion model. This 3D prior, alongside several training strategies, prioritizes the geometry consistency but compromises the texture fidelity. We further propose *Bootstrapped Score Distillation* to specifically boost the texture. We train a personalized diffusion model, Dreambooth, on the augmented renderings of the scene, imbuing it with 3D knowledge of the scene being optimized. The score distillation from this 3D-aware diffusion prior provides view-consistent guidance for the scene. Notably, through an alternating optimization of the diffusion prior and 3D scene representation, we achieve mutually reinforcing improvements: the optimized 3D scene aids in training the scene-specific diffusion model, which offers increasingly view-consistent guidance for 3D optimization. The optimization is thus bootstrapped and leads to substantial texture boosting. With tailored 3D priors throughout the hierarchical generation, DreamCraft3D generates coherent 3D objects with photorealistic renderings, advancing the state-of-the-art in 3D content generation. Code available at https://github.com/deepseek-ai/DreamCraft3D.

## 1 INTRODUCTION

The remarkable success of 2D generative modeling (Saharia et al., 2022; Ramesh et al., 2022; Rombach et al., 2022; Gu et al., 2022) has profoundly shaped the way that we create visual content. 3D content creation, which is crucial for applications like games, movies and virtual reality, still presents a significant challenge for deep generative networks. While 3D generative modeling has shown compelling results for certain categories (Wang et al., 2023a; Chan et al., 2022; Zhang et al., 2023b), generating general 3D objects remains formidable due to the lack of extensive 3D data. Recent research effort has sought to leverage the guidance of pretrained text-to-image generative models (Poole et al., 2022; Lin et al., 2023; Tang et al., 2023) and showcases promising results.

The idea of leveraging pretrained text-to-image (T2I) models for 3D generation is initially proposed by DreamFusion (Poole et al., 2022). A score distillation sampling (SDS) loss is enforced to optimize the 3D model such that its renderings at random viewpoints match the text-conditioned image distribution as interpreted by a powerful T2I diffusion model. DreamFusion inherits the imaginative power of 2D generative models and can yield highly creative 3D assets. To deal with the oversaturation and blurriness issues, recent works adopt stage-wise optimization strategies (Lin et al., 2023) or propose improved 2D distillation loss (Wang et al., 2023b), which leads to an enhancement in photo-realism. However, the majority of current research falls short of synthesizing complex content as achieved by 2D generative models. In addition, these works are often plagued with the "Janus issue", where 3D renderings that appear plausible individually show semantic and stylistic inconsistencies when examined holistically.

---

[*]Work partially done during the internship at DeepSeek AI.
[†]Corresponding authors.

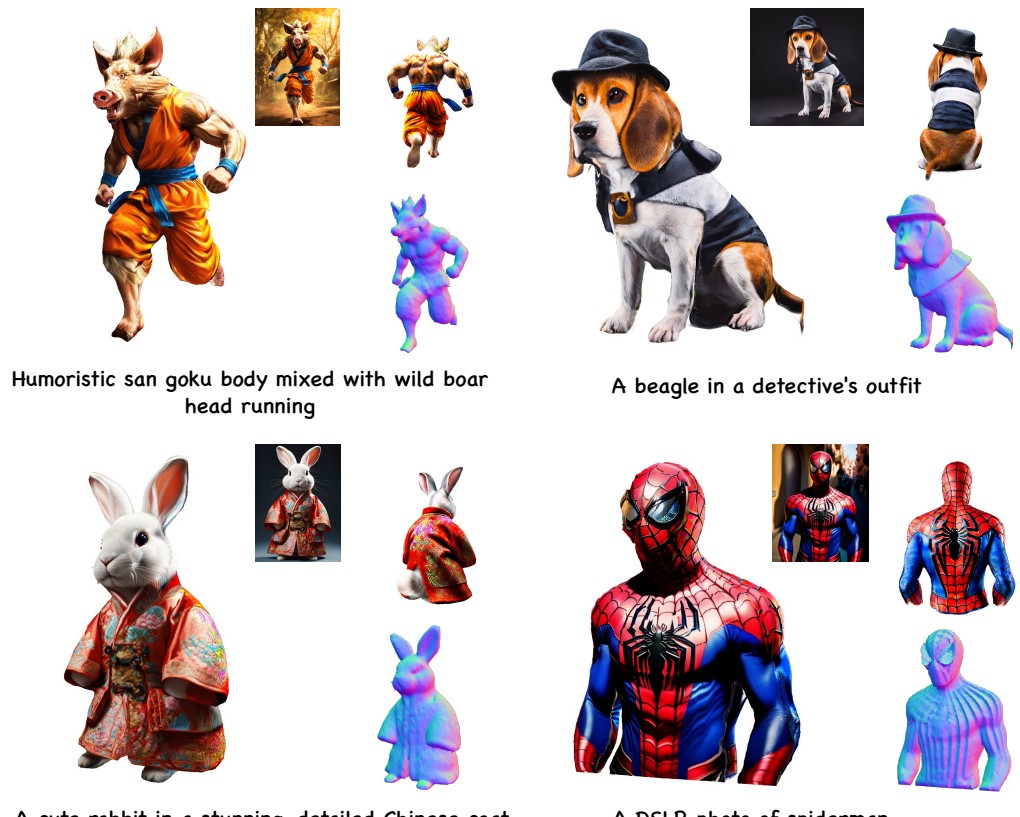

Figure 1: By lifting 2D images to 3D, DreamCraft3D achieves 3D generation with rich details and holistic 3D consistency. Please refer to the Appendix and the demo video for more results.

In this paper, we propose *DreamCraft3D*, an approach to produce complex 3D assets while maintaining holistic 3D consistency. Our approach explores the potential of hierarchical generation. We draw inspiration from the manual artistic process: an abstract concept is first solidified into a 2D draft, followed by the sculpting of rough geometry, the refinement of the geometric details and the painting of high-fidelity textures. We adopt a similar approach, breaking down the challenging 3D generation into manageable steps. Starting with a high-quality 2D reference image generated from a text prompt, we lift it into 3D via stages of geometry sculpting and texture boosting. Contrary to prior approaches, our work highlights how careful consideration of each stage can unleash the full potential of hierarchical generation, resulting in superior-quality 3D creation.

The geometry sculpting stage aims to produce plausible and consistent 3D geometry from the 2D reference image. On top of using the SDS loss for novel views and photometric loss at the reference view, we introduce multiple strategies to promote geometric consistency. Foremost, we leverage an off-the-shelf viewpoint-conditioned image translation model, Zero-1-to-3 (Liu et al., 2023c), to model the distribution of novel views based on the reference image. Since this view-conditioned diffusion model is trained on diverse 3D data (Deitke et al., 2023), it provides a rich 3D prior that complements the 2D diffusion prior. Additionally, we find annealing the sampling timestep and progressively enlarging training views are crucial to further improve coherency. During optimization, we transition from implicit surface representation (Wang et al., 2021) to mesh representation (Shen et al., 2021) for coarse-to-fine geometry refinement. Through these techniques, the geometry sculpting stage produces sharp, detailed geometry while effectively suppressing most geometric artifacts.

We further propose *Bootstrapped Score Distillation* to substantially boost the texture. Existing view-conditioned diffusion models trained on limited 3D often struggle to match the fidelity of modern 2D diffusion models. Instead, we finetune the diffusion model according to multi-view renderings of the 3D instance being optimized. This personalized 3D-aware generative prior becomes instrumental in augmenting the 3D texture while ensuring view consistency. Importantly, we find that

alternately optimizing the generative prior and 3D representation leads to mutually reinforcing improvements. The diffusion model benefits from training on improved multi-view renderings, which in turn provides superior guidance for optimizing the 3D texture. In contrast to prior works (Poole et al., 2022; Wang et al., 2023b) that distill from a fixed target distribution, we learn from a distribution that gradually evolves according to the optimization state. Through this "bootstrapping", our approach captures increasingly detailed texture while keeping the view consistency.

As shown in Figure 1, our method is capable of producing creative 3D assets with intricate geometric structures and realistic textures rendered coherently in 360°. Compared to optimization-based approaches (Poole et al., 2022; Lin et al., 2023), our method offers substantially improved texture and complexity. Meanwhile, compared to image-to-3D techniques (Tang et al., 2023; Qian et al., 2023), our work excels at producing unprecedentedly realistic renderings in 360° renderings. These results suggest the strong potential of DreamCraft3D in enabling new creative possibilities in 3D content creation. The full implementation will be made publicly available.

## 2 RELATED WORK

**3D generative models** have been intensively studied to generate 3D assets without tedious manual creation. Generative adversarial networks (GANs) (Chan et al., 2021; 2022; 2021; Xie et al., 2021; Zeng et al., 2022; Skorokhodov et al., 2023; Gao et al., 2022; Tang et al., 2022; Xie et al., 2021; Sun et al., 2023; 2022) have long been the prominent techniques in the field. Auto-regressive models have been explored (Sanghi et al., 2022; Mittal et al., 2022; Yan et al., 2022; Zhang et al., 2022; Yu et al., 2023), which learn the distribution of these 3D shapes conditioned on images or texts. Diffusion models (Wang et al., 2023a; Cheng et al., 2023; Li et al., 2023; Nam et al., 2022; Zhang et al., 2023a; Nichol et al., 2022; Jun & Nichol, 2023; Bautista et al., 2022; Gupta et al., 2023; Long et al., 2023; Zou et al., 2023; Liu et al., 2023d; Hong et al., 2023) have also shown significant recent success in learning probabilistic mappings from text or images to 3D shape latent. However, these methods require 3D shapes or multi-view data for training, raising challenges when generating in-the-wild 3D assets due to the scarcity of diverse 3D data (Chang et al., 2015; Deitke et al., 2023; Wu et al., 2023) compared to 2D.

**3D-aware image generation** aims to render images in novel views while offering some level of 3D consistency. These works (Sargent et al., 2023; Skorokhodov et al., 2023; Xiang et al., 2023) often rely on a pretrained monocular depth prediction model to synthesize view-consistent images. While some models achieve photo-realistic renderings for ImageNet categories, they struggle with large views. Recent attempts (Watson et al., 2022; Liu et al., 2023c) training view-dependent diffusion models on 3D data show promising results for open-domain novel view synthesis but, as inherently 2D models, can't ensure perfect view consistency.

**Lifting 2D to 3D** approaches improve a 3D scene representation by seeking guidance using estblished 2D text-image foundation models. Early works (Jain et al., 2022; Lee & Chang, 2022; Hong et al., 2022) utilize the pretrained CLIP (Radford et al., 2021) model to maximize the similarity between rendered images and text prompt. DreamFusion (Poole et al., 2022) and SJC (Wang et al., 2022), on the other hand, propose to distill the score of image distribution from a pretrained diffusion model and demonstrate promising results. Recent works have sought to further enhance the texture realism via coarse-to-fine optimization (Lin et al., 2023; Chen et al., 2023), improved distillation loss (Wang et al., 2023b; Liu et al., 2023e; Huang et al., 2023c), shape guidance (Metzer et al., 2023) or lifting 2D image to 3D (Deng et al., 2023; Tang et al., 2023; Qian et al., 2023; Liu et al., 2023b; Huang et al., 2023a). Recently, Raj et al. (2023) proposes to finetune a personalized diffusion model for 3D consistent generation. However, producing globally consistent 3D remains challenging. In this work, we meticulously design 3D priors through the whole hierarchical generation process, achieving unprecedented coherent 3D generation.

## 3 PRELIMINARIES

DreamFusion (Poole et al., 2022) achieves text-to-3D generation by utilizing a pretrained text-to-image diffusion model $\epsilon_\phi$ as an image prior to optimizing the 3D representation parameterized by $\theta$. The image $x = g(\theta)$, rendered at random viewpoints by a volumetric renderer, is expected to represent a sample drawn from the text-conditioned image distribution $p(x|y)$ modeled by a pre-

trained diffusion model. The diffusion model $\phi$ is trained to predict the sampled noise $\boldsymbol{\epsilon}_\phi(\boldsymbol{x}_t; y, t)$ of the noisy image $\boldsymbol{x}_t$ at the noise level $t$, conditioned on the text prompt $y$. A *score distillation sampling* (SDS) loss encourages the rendered images to match the distribution modeled by the diffusion model. Specifically, the SDS loss computes the gradient:

$$\nabla_\theta \mathcal{L}_{\text{SDS}}(\phi, g(\theta)) = \mathbb{E}_{t,\boldsymbol{\epsilon}}\left[\omega(t)(\boldsymbol{\epsilon}_\phi(\boldsymbol{x}_t; y, t) - \boldsymbol{\epsilon})\frac{\partial \boldsymbol{x}}{\partial \theta}\right], \tag{1}$$

which is the per-pixel difference between the predicted and the added noise upon the rendered image, where $\omega(t)$ is the weighting function.

One way to improve the generation quality of a conditional diffusion model is to use the classifier-free guidance (CFG) technique to steer the sampling slightly away from the unconditional sampling, *i.e.*, $\epsilon_\phi(\boldsymbol{x}_t; y, t) + \omega\epsilon_\phi(\boldsymbol{x}_t; y, t) - \omega\epsilon_\phi(\boldsymbol{x}_t, t, \varnothing)$, where $\varnothing$ represents the "empty" text prompt. Typically, the SDS loss requires a large CFG guidance weight for high-quality text-to-3D generation, yet this will bring side effects like over-saturation and over-smoothing (Poole et al., 2022).

Recently, Wang et al. (2023b) proposed a variational score distillation (VSD) loss that is friendly to standard CFG guidance strength and better resolves unnatural textures. Instead of seeking a single data point, this approach regards the solution corresponding to a text prompt as a random variable. Specifically, VSD optimizes a distribution $q^\mu(\boldsymbol{x}_0|y)$ of the possible 3D representations $\mu(\theta|y)$ corresponding to the text $y$, to be closely aligned with the distribution defined by the diffusion timestep $t = 0$, $p(\boldsymbol{x}_0|y)$, in terms of KL divergence:

$$\mathcal{L}_{\text{VSD}} = D_{\text{KL}}(q^\mu(\boldsymbol{x}_0|y)||p(\boldsymbol{x}_0|y)). \tag{2}$$

Wang et al. (2023b) further shows that this objective can be optimized by matching the score of noisy real images and that of noisy rendered images at each time $t$, so the gradient of $\mathcal{L}_{\text{VSD}}$ is

$$\nabla_\theta \mathcal{L}_{\text{VSD}}(\phi, g(\theta)) = \mathbb{E}_{t,\boldsymbol{\epsilon}}\left[\omega(t)(\boldsymbol{\epsilon}_\phi(\boldsymbol{x}_t; y, t) - \boldsymbol{\epsilon}_{\text{lora}}(\boldsymbol{x}_t; y, t, c))\frac{\partial \boldsymbol{x}}{\partial \theta}\right]. \tag{3}$$

Here, $\boldsymbol{\epsilon}_{\text{lora}}$ estimates the score of the rendered images using a LoRA (Low-rank adaptation) (Hu et al., 2021) model. The obtained variational distribution yields samples with high-fidelity textures. However, this loss is applied for texture enhancement and is helpless to the coarse geometry initially learned by SDS. Moreover, both the SDS and VSD attempt to distill from a fixed target 2D distribution which only assures per-view plausibility rather than a global 3D consistency. Consequently, they suffer from the same appearance and semantic shift issue that hampers the perceived 3D quality.

## 4 DREAMCRAFT3D

We propose a hierarchical pipeline for 3D content generation as illustrated in Figure 2. Our method first leverages a state-of-the-art text-to-image generative model to generate a high-quality 2D image from a text prompt. In this way, we can leverage the full power of state-of-the-art 2D diffusion models to depict intricate visual semantics described in the text, retaining the creative freedom of 2D models. We then lift this image to 3D through cascaded stages of geometric sculpting and texture boosting. By decomposing the problem, we can apply specialized techniques at each stage. For geometry, we prioritize multi-view consistency and global 3D structure, allowing for some compromise on detailed textures. With the geometry fixed, we then focus solely on optimizing realistic and coherent texture, for which we jointly learn a 3D-aware diffusion prior that bootstraps the 3D optimization. Next, we elaborate on key design considerations for the two phases.

### 4.1 GEOMETRY SCULPTING

At this stage, we aim to craft a 3D model such that it matches the appearance of the reference image $\hat{\boldsymbol{x}}$ at the same reference view while maintaining plausibility under different viewing angles. To achieve this, we encourage plausible image renderings for each randomly sampled view, recognizable by a pretrained diffusion model. This is achieved using the SDS loss $\mathcal{L}_{\text{SDS}}$, as defined in Equation 1. In order to effectively utilize guidance from the reference image, we penalize the photometric difference between the rendered image and the reference via $\mathcal{L}_{\text{rgb}} = \|\hat{\boldsymbol{m}} \odot (\hat{\boldsymbol{x}} - g(\theta; \hat{c}))\|_2$ at the reference view $\hat{c}$. The loss is computed only within the foreground region denoted by the mask $\hat{\boldsymbol{m}}$. Meanwhile, we implement the mask loss $\mathcal{L}_{\text{mask}} = \|\hat{\boldsymbol{m}} - g_m(\theta; \hat{c})\|_2$ to encourage scene

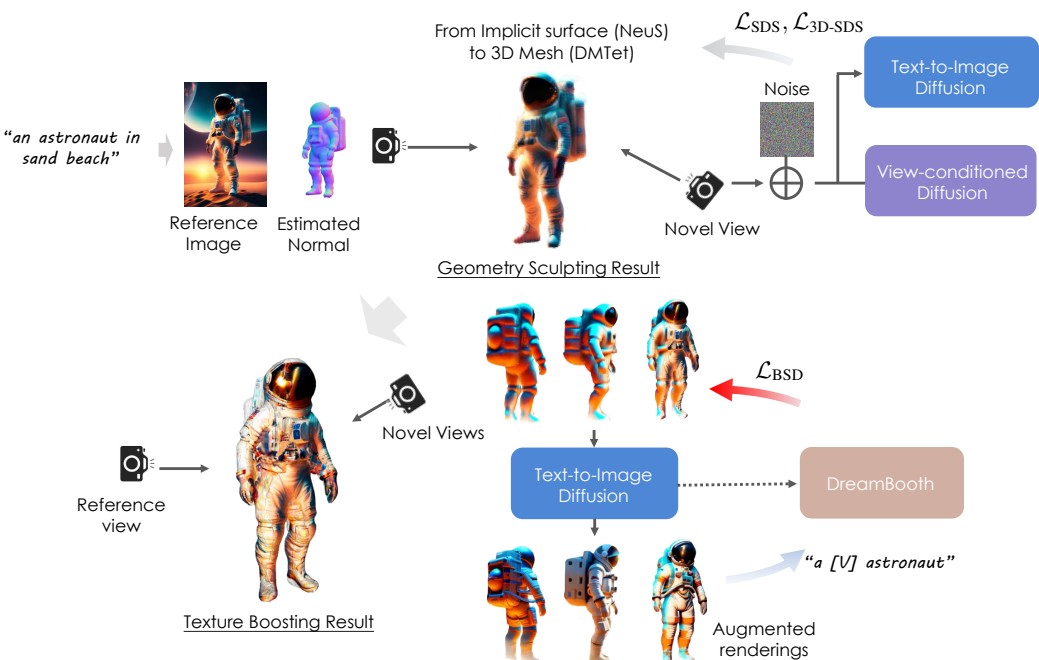

Figure 2: DreamCraft3D leverages a 2D image generated from the text prompt and uses it to guide the stages of geometry sculpting and texture boosting. When sculpting the geometry, the view-conditioned diffusion model provides crucial 3D guidance to ensure geometric consistency. We then dedicately improve the texture quality by conducting a cyclic optimization. We augment the multi-view renderings and use them to finetune a diffusion model, DreamBooth, to offer multi-view consistent gradients to optimize the scene. We term the loss that distills from an evolving diffusion prior as bootstrapped distillation sampling ($\mathcal{L}_{\text{BSD}}$ in the figure).

sparsity, where $g_m$ renders the silhouette. In addition, akin to (Deng et al., 2023), we fully exploit the geometry prior inferred from the reference image, and enforce the consistency with the depth and normal map computed for the reference view. The corresponding depth and normal loss are respectively computed as:

$$\mathcal{L}_{\text{depth}} = -\frac{\text{conv}(\boldsymbol{d}, \hat{\boldsymbol{d}})}{\sigma(\boldsymbol{d})\sigma(\hat{\boldsymbol{d}})}, \quad \mathcal{L}_{\text{normal}} = -\frac{\boldsymbol{n} \cdot \hat{\boldsymbol{n}}}{\|\boldsymbol{n}\|_2 \cdot \|\hat{\boldsymbol{n}}\|_2}, \tag{4}$$

where $\text{conv}(\cdot)$ and $\sigma(\cdot)$ represent the covariance and variance operators respectively, and the depth $\hat{\boldsymbol{d}}$ and the normal $\hat{\boldsymbol{n}}$ at the reference view are computed using the off-the-shelf single-view estimator (Eftekhar et al., 2021). The depth loss adopts the form of negative Pearson correlation $\mathcal{L}_{\text{depth}}$ to account for the scale mismatch in depth.

Despite these, maintaining consistent semantics and appearance across back-views remains a challenge. Thus, we employ additional techniques to produce coherent, detailed geometry.

**3D-aware diffusion prior.** We argue that the 3D optimization with per-view supervision alone is under-constrained. Hence, we utilize a view-conditioned diffusion model, Zero-1-to-3, which is trained on a large scale of 3D assets and offers an improved viewpoint awareness. The Zero-1-to-3 is a fine-tuned 2D diffusion model, which hallucinates the image in a relative camera pose $c$ given the reference image $\hat{\boldsymbol{x}}$. This 3D-aware model encodes richer 3D knowledge of the visual world and allows us to better extrapolate the views given a reference image. As such, we distill the probability density from this model and compute the gradient of a 3D-aware SDS loss for novel views:

$$\nabla_\theta \mathcal{L}_{\text{3D-SDS}}(\phi, g(\theta)) = \mathbb{E}_{t,\epsilon}[\omega(t)(\epsilon_\phi(\boldsymbol{x}_t; \hat{\boldsymbol{x}}, c, y, t) - \epsilon)\frac{\partial \boldsymbol{x}}{\partial \theta}]. \tag{5}$$

This loss effectively alleviates 3D consistency issues like the Janus problem. However, the fine-tuning on limited categories of 3D data of inferior rendering quality impairs the diffusion model's generation capability, so the 3D-aware SDS loss alone is prone to induce deteriorated quality when

lifting general images to 3D. Therefore, we employ a hybrid SDS loss, which incorporates both the 2D and 3D diffusion priors simultaneously. Formally, this hybrid SDS loss provides the gradient as:

$$\nabla_\theta \mathcal{L}_{\text{hybrid}}(\phi, g(\theta)) = \nabla_\theta \mathcal{L}_{\text{SDS}}(\phi, g(\theta)) + \mu \nabla_\theta \mathcal{L}_{\text{3D-SDS}}(\phi, g(\theta)), \tag{6}$$

where we choose $\mu = 2$ to emphasize the weight of the 3D diffusion prior. When computing $\mathcal{L}_{\text{SDS}}$, we adopt the DeepFloyd IF base model (Shonenkov et al., 2023), a diffusion model that operates at $64 \times 64$ resolution pixel space and better captures coarse geometry.

**Diffusion timestep annealing.** To align with the coarse-to-fine progression of 3D optimization, we adopt a diffusion timestep annealing strategy similar to Huang et al. (2023b). At the start of optimization, we prioritize sampling larger diffusion timestep $t$ from the range $[0.7, 0.85]$ when computing Equation 6 to provide the global structure. As training proceeds, we linearly anneal the $t$ sampling range to $[0.2, 0.5]$ over hundreds of iterations. This annealing strategy allows the model to first establish a plausible global geometry in the early optimization phase before refining the structural details.

**Detailed structural enhancement.** We initially optimize an implicit surface representation with the corresponding volume rendering as in NeuS (Wang et al., 2021) to establish the coarse structure. Then, following Lin et al. (2023), we use this result to initialize a textured 3D mesh representation using a deformable tetrahedral grid (DMTet) (Shen et al., 2021) to facilitate high-resolution details. Moreover, this representation disentangles the learning of geometry and texture. Hence, at the end of this structural enhancement, we are able to solely refine the texture and better preserve high-frequency details from the reference image.

We provide a summary of the total loss in the geometry sculpting stage, denoted as $\mathcal{L}_{\text{gs}}$, as follows:

$$\mathcal{L}_{\text{gs}} = \lambda_{\text{rgb}}\mathcal{L}_{\text{rgb}} + \lambda_{\text{mask}}\mathcal{L}_{\text{mask}} + \lambda_{\text{depth}}\mathcal{L}_{\text{depth}} + \lambda_{\text{normal}}\mathcal{L}_{\text{normal}} + \lambda_{\text{hybrid}}\mathcal{L}_{\text{hybrid}}. \tag{7}$$

We set $\lambda_{\text{rgb}} = 10000$, $\lambda_{\text{mask}} = 5000$, $\lambda_{\text{depth}} = \lambda_{\text{normal}} = 0.1$, $\lambda_{\text{hybrid}} = 1$.

## 4.2 TEXTURE BOOSTING VIA BOOTSTRAPPED SCORE SAMPLING

The geometry sculpting stage prioritizes the learning of coherent and detailed geometry but leaves the texture blurry. This is due to our reliance on a 2D prior model that operates at a coarse resolution, and the limited sharpness offered by the 3D-aware diffusion model. Additionally, texture issues such as over-smoothing and over-saturation arise from excessively large classifier-free guidance.

To augment the texture realism, we use variational score distillation (VSD) loss, as detailed in Equation 3. We switch to the Stable Diffusion model (Rombach et al., 2021) in this stage which offers high-resolution gradients. To promote realistic rendering, we exclusively optimize the mesh texture with the tetrahedral grid fixed. In this learning stage, we do not leverage the Zero-1-to-3 model as the 3D prior since it adversely impacts the texture quality. Nonetheless, the inconsistent textures may come back, resulting in bizarre 3D outcomes.

We observe that the multi-view renderings from the previous stage, despite some blurriness, exhibit good 3D consistency. One idea is to adapt a pretrained 2D diffusion model using these rendering results, enabling the model to form a concept about the scene's surrounding views. In light of this, we finetune the diffusion model with the multi-view image renderings $\{x\}$, using DreamBooth (Ruiz et al., 2023). Specifically, we incorporate the text prompts containing a unique identifier and the subject's class name (e.g., "A [V] astronaut" in Figure 2). During finetuning, the camera parameter of each view is introduced as an additional condition. In practice, we train the DreamBooth with "augmented" image renderings, $x_r = r_{t'}(x)$. We introduce Gaussian noises, in an amount specified by the diffusion timestep $t'$, to the multi-view renderings, i.e., $x_{t'} = \alpha_{t'}x_0 + \sigma_{t'}\epsilon$ ($\alpha_{t'}, \sigma_{t'} > 0$ are hyperparameters), which are restored using the diffusion model. Choosing a large $t'$ reveals high-frequency details in augmented images at the expense of fidelity to original renderings. The DreamBooth model trained on these renderings can guide texture refinement as a 3D prior.

Further, we propose to alternatively optimize the 3D scene to facilitate a bootstrapped optimization (Figure 2). Initially, the 3D mesh yields blurry multi-view renderings. We adopt a large diffusion $t'$ to augment their texture quality while introducing some 3D inconsistency. The DreamBooth model trained on these augmented renderings obtains a unified 3D concept of the scene to guide texture

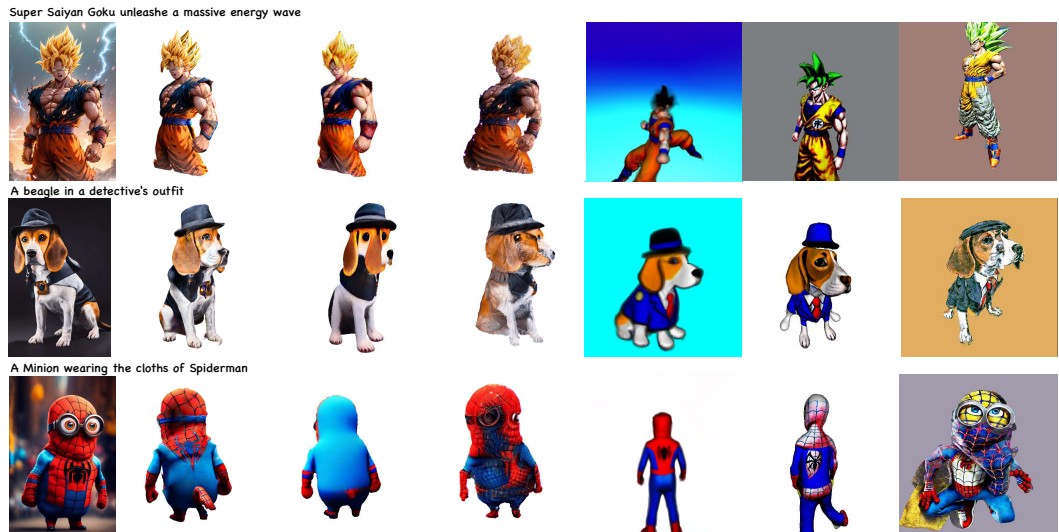

Super Saiyan Goku unleashe a massive energy wave

A beagle in a detective's outfit

A Minion wearing the cloths of Spiderman

| Reference | Ours | Magic123 | Make-It-3D | DreamFusion | Magic3D | ProlificDreamer |

Figure 3: Qualitative comparison with baselines. Our method generates sharper and more plausible details in both geometry and texture. Note that our method generates rich texture detail at novel views and eliminates multi-face Janus problems.

refinement. As the 3D mesh reveals finer textures, we reduce the diffusion noises introduced to the image renderings, so the DreamBooth model learns from more consistent renderings and better captures the image distribution faithful to evolving views. In this cyclic process, the 3D mesh and diffusion prior mutually improve in a bootstrapped manner. Formally, we derive the 3D optimization gradient using the following bootstrapped score distillation (BSD) loss:

$$\nabla_\theta \mathcal{L}_{\text{BSD}}(\phi, g(\theta)) = \mathbb{E}_{t,\epsilon,c}[\omega(t)(\boldsymbol{\epsilon}_{\text{DreamBooth}}(\boldsymbol{x}_t; y, t, \{\tilde{x}\}_k, c) - \boldsymbol{\epsilon}_{\text{lora}}(\boldsymbol{x}_t; y, t, \{x\}, c))\frac{\partial \boldsymbol{x}}{\partial \theta}]. \quad (8)$$

The notation $\{\tilde{x}\}_k$ denotes the augmented renderings generated during the $k^{th}$ iteration of Dream-Booth. Contrary to prior works (Poole et al., 2022; Wang et al., 2023b) that distill the score function from a fixed 2D model, our BSD loss learns from an evolving model which becomes increasingly 3D consistent by drawing feedback from the ongoing crafted 3D model. In our experiments, we alternate the optimization twice, which suffices to produce consistent textures with rich details.

The total loss in the texture stage, labeled as $\mathcal{L}_{\text{texture}}$, can be expressed as below:

$$\mathcal{L}_{\text{texture}} = \lambda_{\text{rgb}}\mathcal{L}_{\text{rgb}} + \lambda_{\text{BSD}}\mathcal{L}_{\text{BSD}}. \quad (9)$$

We set $\lambda_{\text{rgb}} = 10000$ and $\lambda_{\text{BSD}} = 1$, values empirically suitable for most cases.

## 5 EXPERIMENTS

### 5.1 IMPLEMENTATION DETAILS

**Architectural details.** In the geometry sculpting stage, we use NeuS and textured 3D mesh representations. We employ Instant NGP (Müller et al., 2022), optimizing from a 64 to a 384 resolution. For the textured mesh, we use DMTet at a 128 grid and 512 rendering resolution.

**Optimization.** In mesh refinement, given a plausible global geometric structure, we skip the use of a 3D prior in texture optimization. We employ random camera radius and FOV angle sampling, alternately rendering normal maps and RGB images to enhance texture and geometry.

### 5.2 COMPARISONS WITH THE STATE OF THE ARTS

**Baselines.** We conduct a comparative analysis of our technique against five baseline methods. The first three are text-to-3D methods: DreamFusion (Poole et al., 2022), Magic3D (Lin et al., 2023) and

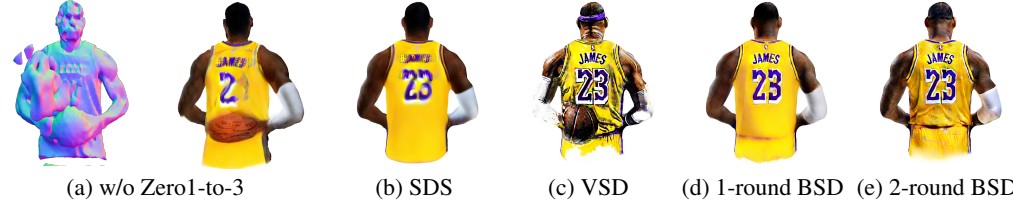

(a) w/o Zero1-to-3     (b) SDS     (c) VSD     (d) 1-round BSD   (e) 2-round BSD

Figure 4: Ablation study of the effectiveness of 3D prior and our proposed BSD (Bootstrapped Score Distillation).(a) Geometry sculpting stage without 3D prior. (b) Texture optimization with SDS loss. (c) VSD loss leads to richer texture detail but inconsistent texture. (d) BSD enhances texture consistency with one DreamBooth round. (e) Two-round BSD adds more detail to the result.

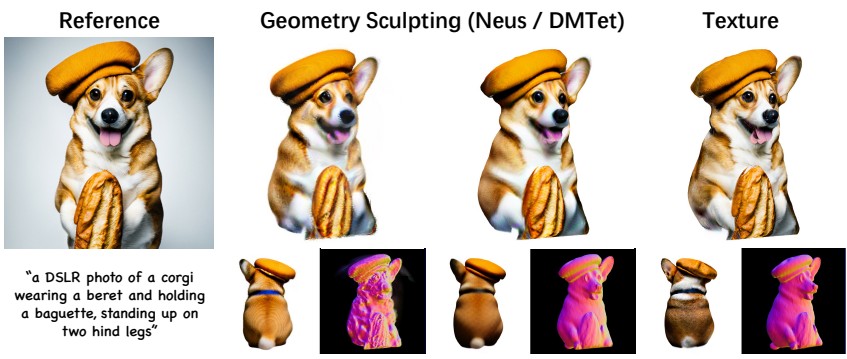

Figure 5: Continual improvement of geometry and texture quality through multiple stages.

ProlificDreamer (Wang et al., 2023b). We also compare our method against two image-to-3d methods: Make-it-3D (Tang et al., 2023) and Magic123 (Qian et al., 2023). For DreamFusion, Magic3D, Magic123, and ProlificDreamer, we utilize their implementations in the Threestudio library (Guo et al., 2023) for comparison. For Make-it-3D, we use its official implementation.

**Datasets.** We establish a test benchmark that includes 300 images, which is a mix of real pictures and those produced by Stable Diffusion (Rombach et al., 2021) and Deep Floyd. Each image in this benchmark comes with an alpha mask for the foreground, a predicted depth map, and a text prompt. For real images, the text prompts are sourced from an image caption model. We intend to make this test benchmark accessible to the public.

Table 1: Quantitative comparison against prior 2D-to-3D lifting methods. The metrics are measured on 300 generated samples.

|  | CLIP ↑ | Contextual ↓ | PSNR ↑ | LPIPS ↓ | Training Time |
|---|---|---|---|---|---|
| Make-it-3D | 0.872 | 1.609 | 18.937 | 0.054 | ∼1.5h |
| Magic123 | 0.843 | 1.628 | 22.838 | 0.053 | ∼2.8h |
| **DreamCraft3D** | **0.896** | **1.579** | **31.801** | **0.005** | ∼2.3h |

**Quantitative comparison.** To generate compelling 3D content that resembles the input image and consistently conveys semantics from various perspectives, we compare our technique with established baselines using a quantitative analysis. Our evaluation employed four metrics: LPIPS (Zhang et al., 2018) and PSNR for fidelity measurement at the reference viewpoint; Contextual Distance (Mechrez et al., 2018) and CLIP score (Radford et al., 2021) to estimate pixel and semantic-level coherence respectively. Table 1 indicates that our approach significantly surpasses the baselines in maintaining both texture consistency and fidelity. Regarding training efficiency, our method requires approximately 2.3 hours for each case, which is comparable to two other methods.

**Qualitative comparison.** Figure 3 compares our method with the baselines. All the text-to-3D methods suffer from multi-view consistency issues. While ProlificDreamer offers realistic textures, it fails to form a plausible 3D object. Image-to-3D methods like Make-it-3D create quality frontal views but struggle with geometry. Magic123, enhanced by Zero1-to-3, fares better in geometry

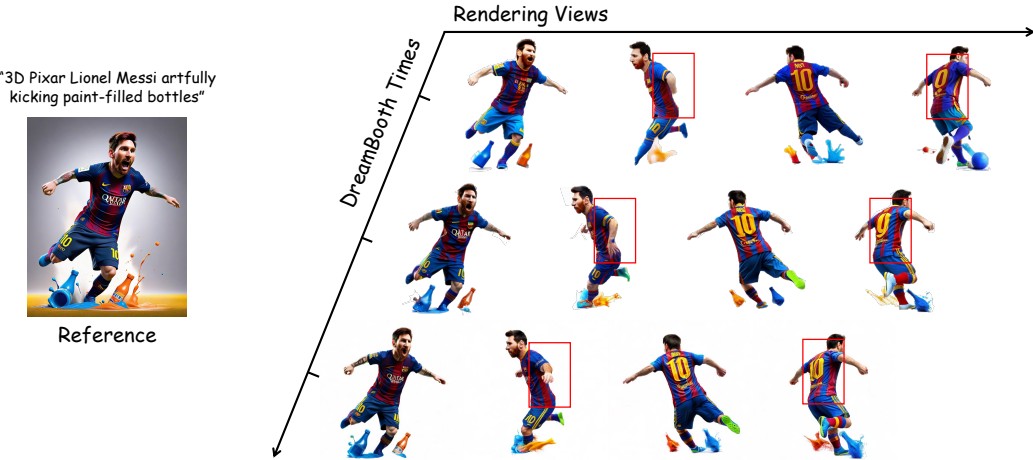

Figure 6: Improved view consistency and texture fidelity along bootstrapping.

regularization, but both generate overly smoothed textures and geometric details. In contrast, our BSD improves imagination diversity while maintaining semantic consistency.

## 5.3 ANALYSIS

**The effect of 3D prior.** In our paper we claim that the guidance offered by a 3D prior enhances the generation of globally plausible geometry. To ascertain its impact, an ablation study is conducted, where the 3D prior is deactivated. Figure 4 shows that without a 3D prior, the character tends to display the Janus issue and irregular geometry, highlighting the crucial role of a viewpoint-aware 3D prior in maintaining global shape consistency.

**The effect of BSD.** Figure 4 also presents an ablation study encompassing three texture optimization techniques: (1) BSD, (2) VSD, and (3) Score Distillation Sampling (SDS) with the traditional stable diffusion. The application of SDS has been observed to generate novel-view textures that are excessively smooth and over-saturated. In contrast, while VSD using standard stable diffusion can produce realistic textures, it yields a notably high inconsistency. In contrast, our proposed approach successfully generates textures that strike a balance between realism and consistency.

**Visualization of multiple stages.** Figure 5 visualizes the intermediate rendering results for each stage in our hierarchical pipeline. In the geometry sculpting stage, we convert NeuS to DMTet to improve high-resolution geometry details. However, the improvement in texture is negligible. On the contrary, in the texture stage, we significantly improve the texture quality with BSD.

**DreamBooth times.** Figure 6 shows multi-view datasets for DreamBooth. Initially, noise amplifies detail richness but leads to inconsistent denoised images. As optimization proceeds, renderings evolve towards greater consistency and photorealism, enhancing the DreamBooth-tailored input dataset's quality.

## 6 CONCLUSION

We have presented DreamCraft3D, an innovative approach that advances the field of complex 3D asset generation. This work introduces a meticulous geometry sculpting phase for producing plausible and coherent 3D geometries and a novel Bootstrapped Score Distillation strategy. The latter, by distilling from an optimizing 3D-aware diffusion prior and adapting to multi-view renderings of the instance being optimized, significantly improves texture quality and consistency. DreamCraft3D produces high-fidelity 3D assets with compelling texture details and multi-view consistency. We believe this work represents an important step towards democratizing 3D content creation and shows great promise in future applications.

ACKNOWLEDGMENTS

This work was supported by the Natural Science Foundation of China under Grants No. 61827805 and No. 62125107.

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

## A  APPENDIX

### A.1  IMPLEMENTATION DETAILS

**Algorithms for Bootstrapped Score Distillation.** We provide a summarized algorithm of bootstrapped score distillation in Algorithm 1. The "Bootstrapped Score Distillation" algorithm starts by initializing $n$ ($n = 1$ in our case) meshes and a pretrained text-to-image diffusion model $\epsilon_{\text{DreamBooth}}$ parameterized by $\phi$. The algorithm then enters an iterative loop: in each iteration, it renders the mesh to obtain multi-view images $x$, augments these images with Gaussian noises to form $x_{t'} = \alpha_{t'} x + \sigma_{t'} \epsilon$, and fine-tunes the pretrained diffusion model $\epsilon_{\text{DreamBooth}}$ based on these augmented image renderings. Within each iteration, there is an inner loop running for $n$ steps, where a random mesh and camera pose are sampled, and a 2D image is rendered from the chosen pose. Then, updates are performed on $\theta$ and $\phi$ using gradients calculated from the difference between the pretrained score function and the predicted score function, and from the $\mathcal{L}_2$ norm between the predicted score and real noise, respectively. These iterations continue until convergence, and the final refined mesh(s) are returned.

**Structure-aware latent regularization.** To maintain the high-quality output produced by BSD while reducing noise and inconsistencies, we further incorporate a control net-guided inpainting diffusion model that regularizes the generated textures. Specifically, for a rendered image x from an arbitrary viewpoint, the visible section under the reference view is initially computed. This invariant portion during the generation process allows our inpainting model to fill in the remaining segments. As these remaining parts adhere to geometric constraints, we integrate geometric normal information through a control net. Ultimately, this method permits us to enforce view consistency and generate realistic results using a ControlNet-guided inpainting diffusion model. To preserve the high-quality generation output, we avoid utilizing this image directly as a loss against the rendered image. Instead, we subtly introduce it by constraining the norm of the latent variables:

$$\mathcal{L}_{\text{reg}}(\phi, g(\theta)) = \Sigma(\|E(x)\|_2 - \|E(x_{\text{reg}})\|_2)^2. \tag{10}$$

**Architectural details.** In the NeuS approach (Wang et al., 2021), we employ a single-layer Multi-Layer Perceptron (MLP) with 32 hidden units to simultaneously predict RGB color, volume density, and normal. The inputs to this MLP are the concatenated feature vectors derived from multi-resolution hash encoding sampled with trilinear interpolation. To sparsify the Instant NGP representation, we implement density-based pruning every 10 iterations within an octree structure, as suggested by Magic3D (Lin et al., 2023). In our experiments, we use a bounding sphere with a radius of 2. For the density prediction, we utilize the softplus activation function and, following the approach of Poole et al. 2022, include an initial spatial density bias to encourage optimization in favor of the object-centric neural field.

**Camera and light augmentations.** We follow Magic3D to add random augmentations to the camera and light sampling for rendering the shaded images. Differently, we sample the point light location such that the angular distance from the random camera center location (w.r.t. the origin) is sampled with a random point light distance $r_{\text{cam}} \sim (7.5, 10)$, and (b) we freeze the material augmentation unlike Dreamfusion and Magic3D, as we found it is bad for training convergence (c) In the coarse stage, we propose a fixed-random mixed camera pose strategy. Specifically, following the common practice of the current text-to-3D methods, random camera view sampling benefits scene optimization. However, Zero1-to-3 needs fixed camera intrinsic parameters. Therefore, we let half of the GPUs sample the camera distance from $\mathcal{U}(3.2, 3.5)$, and the Field-of-View from $\mathcal{U}(10, 20)$, while the left GPUs are fixed to the default camera intrinsic.

**Time annealing.** At the beginning of the geometry sculpting stage, We utilize a simple two-stage annealing of time step t in the score distillation objective. For the first several iterations we sample time steps $t \sim \mathcal{U}(0.7, 0.85)$ and then anneal into $t \sim \mathcal{U}(0.2, 0.50)$. We refer the readers to Prolific-Dreamer (Wang et al., 2023b). For the left iterations, we fix time steps to $\mathcal{U}(0.2, 0.50)$. We also utilize a simple two-stage time annealing for the multi-view dataset generation, that is, for the first updating step, we select a time step $t = 0.5$ for all rendered images and then anneal it into $t = 0.1$ along the later updating steps.

**Progressive training.** We linearly increase the sampling range of camera positions with elevation angle ($\phi_{\text{cam}}$) from $0°$ to $[-10°, 45°]$, and azimuth angle ($\theta_{\text{cam}}$) from $0°$ to $[-180°, 180°]$. The progress length is set as 200 iterations.

## A.2 DIVERSE RESULTS

Prior studies frequently yield models of limited diversity with disproportionately smooth textures. Our approach to superior text-to-3D generation initially translates the text prompt into a reference image via 2D diffusion before implementing our proprietary image-based 3D creation methodology. Figure 7 demonstrates the proficiency of our method in generating an array of diverse models from a single text prompt, all characterized by their remarkable quality.

## A.3 LIMITATIONS

Our approach occasionally incorporates frontal-view geometric details into texture, as depicted in Figure 8, due to depth ambiguity and inaccuracies in the depth prior. Furthermore, we do not expressly segregate material and lighting from the 2D reference image, an aspect deferred for future exploration.

---

**Algorithm 1** Bootstrapped Score Distillation

---

**Input:** Number of particles $n$ ($\geq 1$). Pretrained text-to-image diffusion model $\epsilon_{\text{DreamBooth}}$. Learning rate $\eta_1$ and $\eta_2$ for 3D structures and diffusion model parameters, respectively. A prompt $y$. Number of images $m$ and Camera poses $\{c_r^{(i)}\}_{i=1}^m$ for the multi-view datasets.

1: **initialize** $n$ meshes $\{\theta^{(i)}\}_{i=1}^n$, a noise prediction model $\epsilon_\phi$ parameterized by $\phi$.
2: **while** not converged **do**
3:      Render the mesh to get multi-view images $\boldsymbol{x} = g(\theta, c_r)$.
4:      Augment image renderings with Gaussian noises: $\boldsymbol{x}_{t'} = \alpha_{t'}\boldsymbol{x} + \sigma_{t'}\boldsymbol{\epsilon}$.
5:      Finetune $\epsilon_{\text{DreamBooth}}$ on augmented image renderings $\boldsymbol{x_r} = r_{t'}(\boldsymbol{x})$.
6:      **for** i in $T$ steps **do**
7:          Randomly sample $\theta \sim \{\theta^{(i)}\}_{i=1}^n$ and a camera pose $c$.
8:          Render the 3D structure $\theta$ at pose $c$ to get a 2D image $\boldsymbol{x}_0 = \boldsymbol{g}(\theta, c)$.
9:          $\theta \leftarrow \theta - \eta_1 \mathbb{E}_{t,\epsilon,c} \left[ \omega(t) \left( \epsilon_{\text{DreamBooth}}(\mathbf{x}_t, t, c, y) - \epsilon_\phi(\mathbf{x}_t, t, c, y) \right) \frac{\partial g(\theta, c)}{\partial \theta} \right]$
10:          $\phi \leftarrow \phi - \eta_2 \nabla_\phi \mathbb{E}_{t,\epsilon} || \boldsymbol{\epsilon}_\phi(\mathbf{x}_t, t, c, y) - \epsilon ||_2^2$.
11:      **end for**
12: **end while**
13: **return**

---

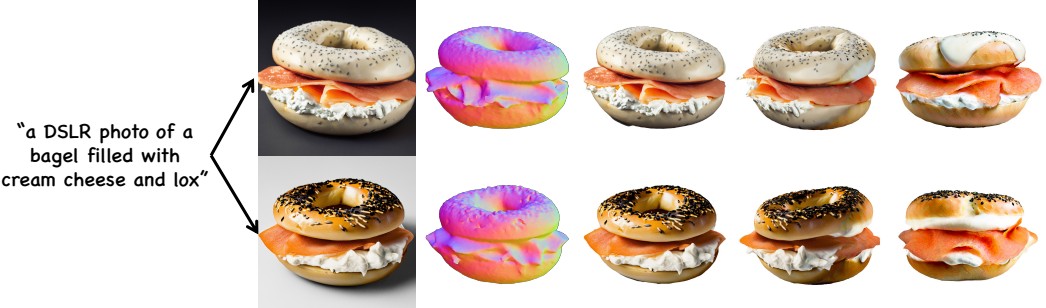

"a DSLR photo of a bagel filled with cream cheese and lox"

Figure 7: DreamCraft3D skillfully generates an assortment of visually compelling 3D models when provided with a textual description.

### A.4 ADDITIONAL RESULTS FOR ABLATION STUDIES

**The effectiveness of BSD.** We conduct quantitative and qualitative experiments to assess the effectiveness of the proposed BSD. Table 2 summarizes the evaluation of 3D generation quality on four metrics: LPIPS (Zhang et al., 2018) and PSNR for fidelity measurement at the reference view; Contextual Distance (Mechrez et al., 2018) and CLIP score (Radford et al., 2021) for pixel and semantic-level coherence respectively. The use of SDS loss in texture refinement results in view-coherent texture but fails to restore high-quality textural details. In contrast, VSD significantly improves the texture generation quality but tends to degrade view consistency on texture. However, when employing BSD, we obtain sharp and view-consistent textured meshes. Furthermore, the results improve with each round of DreamBooth, indicating the effectiveness of the bootstrapping.

Figure 10 illustrates the efficacy of BSD. Consistent with Figure 4, we can conclude that BSD uniquely enables the generation of high-quality texture while preserving style consistency with the reference image.

**The effectiveness of designs in the geometry sculpting stage.** We conduct supplementary ablation studies on various design aspects within the geometry sculpting stage, which include loss functions, training strategies, and model choices. For these tests, we report quantitative and qualitative results for NeuS instead of DMTet to isolate the impact of changing 3D representations.

As depicted in Table 3, disabling $\mathcal{L}_{\text{RGB}}$ significantly drops reconstruction fidelity due to lack of color reinforcement. Training without $\mathcal{L}_{\text{mask}}$ results in less but notable fidelity degradation. Also, omitting 3D prior (Liu et al., 2023c) during training negatively affects global semantic coherence.

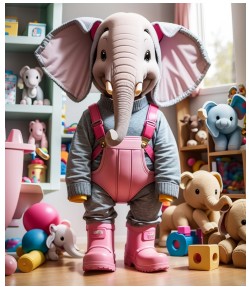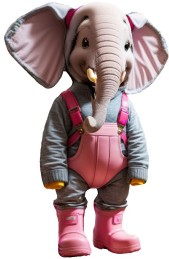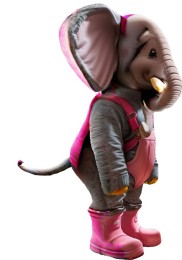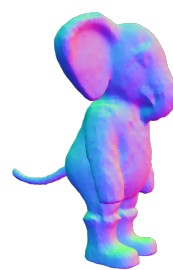

Figure 8: Failure case. Our method learns incorrect geometry for elephant nose.

Table 3 also reveals that omitting either progressive training or diffusion timestep annealing during the geometry sculpting stage leads to a decrease in CLIP and contextual scores, indicating their crucial roles in early geometry modeling optimization.

For the diffusion guidance, replacing DeepFloyd with Stable Diffusion 2.1 (Rombach et al., 2022) in an ablation study leads to decreased generation quality, potentially due to DeepFloyd's superior view-awareness from using T5-XXL (Raffel et al., 2020) as a text encoder. Similarly, DreamFusion (Poole et al., 2022), which also employs the T5-XXL encoder, exhibits fewer Janus issues than its reimplementation with Stable Diffusion (Tang, 2022).

Figure 11 visualizes the ablation of our design choices. Obviously, the default setting yields the highest fidelity and imaginative power in generating 3D contents.

**Training time for individual stages.** In addition to the efficiency comparison with baselines in Table 1, we also provide training times for individual stages, shown in Table 4. Averaged on 300 objects in our test bench, the total training time is about 2.3 hours, of which 1.5 hours are dedicated to the geometry sculpting stage and 0.8 hours to the texture stage. The inference time is approximately 0.05 seconds per frame at a resolution of 512. We conducted our timing tests using 8 A100 GPUs for training and a single A100 GPU for inference.

**3D representation choice.** We choose NeuS (Wang et al., 2021) instead of NeRF (Mildenhall et al., 2021) as the 3D representation in the early geometry sculpting stage. As shown in Figure 12, NeRF learns a volume density field, from which it is difficult to extract a high-quality surface, e.g. hole artifacts, for DMTet initialization. On the contrary, NeuS learns a neural SDF representation, enabling subtle surface extraction. Moreover, the DeepFloyd base model offers gradient on a coarse resolution (64x64), which enforces a coherent 3D geometry rather than introducing high-frequency signals that distract the geometry sculpting.

### A.5 ROBUSTNESS TO OUT-OF-DOMAIN REFERENCE IMAGES

Our method adopts a 3D-aware diffusion prior, Zero123 (Liu et al., 2023c) for guiding optimization in the geometry sculpting stage. Zero123 is trained on a large synthetic 3D dataset, Objaverse (Deitke et al., 2023). In order to prove the robustness of our method on images far from the data distribution of Objaverse, we select 4 reference images from showcases in our paper and use them to retrieve 3D objects within Objaverse. The retrieval algorithm is based on OpenShape (Liu et al., 2023a). As shown in Figure 13, the retrieval results have a significant domain gap with the reference images, while our results show promising 3D generations for these cases. For example, we generate "Messi" with photo-realistic back jersey details from the reference image though there is no similar 3D data in Objaverse.

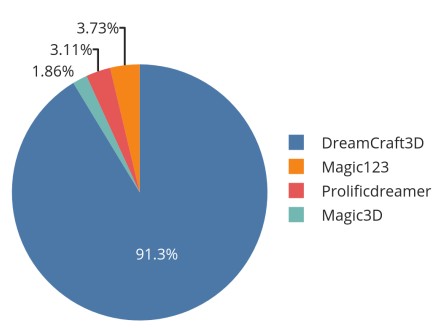

Figure 9: User study.

Table 2: Quantitative ablation study of the effectiveness of proposed BSD (Bootstrapped Score Distillation).

| | CLIP ↑ | Contextual ↓ | PSNR ↑ | LPIPS ↓ |
|---|---|---|---|---|
| SDS | 0.775 | 1.894 | 25.258 | 0.036 |
| VSD | 0.792 | 1.833 | 31.426 | 0.005 |
| One-round BSD | 0.863 | 1.647 | 30.431 | 0.008 |
| Two-round BSD | **0.896** | **1.579** | **31.801** | **0.005** |

Table 3: Quantitative ablation study of the effectiveness of design choices in the geometry sculpting stage.

| | CLIP ↑ | Contextual ↓ | PSNR ↑ | LPIPS ↓ |
|---|---|---|---|---|
| Default setting | **0.745** | **1.936** | **20.836** | 0.031 |
| w/o $\mathcal{L}_{\text{RGB}}$ | 0.619 | 2.212 | 10.562 | 0.095 |
| w/o $\mathcal{L}_{\text{mask}}$ | 0.736 | 1.949 | 19.307 | 0.057 |
| w/o 3D prior | 0.679 | 2.105 | 20.252 | 0.092 |
| w/o time annealing | 0.737 | 1.943 | 20.120 | 0.035 |
| w/o progressive training | 0.721 | 1.962 | 20.317 | **0.029** |
| Stable diffusion | 0.669 | 2.154 | 18.182 | 0.099 |

To substantiate the robustness and quality of our proposed model, we executed a user study employing 15 distinct pairs of prompts and images. Each participant was provided with four free-view rendering videos alongside their corresponding textual inputs and asked to choose their top preferred 3D model. The study gathered 480 responses from a total of 32 participants, the analysis of which is depicted in Figure 9. On an average basis, our model was favored by 92% of users over alternative models, outperforming the baselines by a large margin. This result provides compelling evidence of the resilience and superior quality inherent to our proposed method.

## A.6 ADDITIONAL QUALITATIVE RESULTS

Figure 14−Figure 17 provides more results produced by DreamCraft3D. Our method is able to produce photo-realistic 3D assets with compelling textural details. Moreover, our method shows significantly improved 3D consistency. Please find the video results in the supplementary video.

Table 4: Training and inference time for our method.

|  | Total training time | Geometry sculpting Stage | Texture Stage | Inference |
|---|---|---|---|---|
| DreamCraft3D | 2.3h | 1.5h | 0.8h | 0.05s |

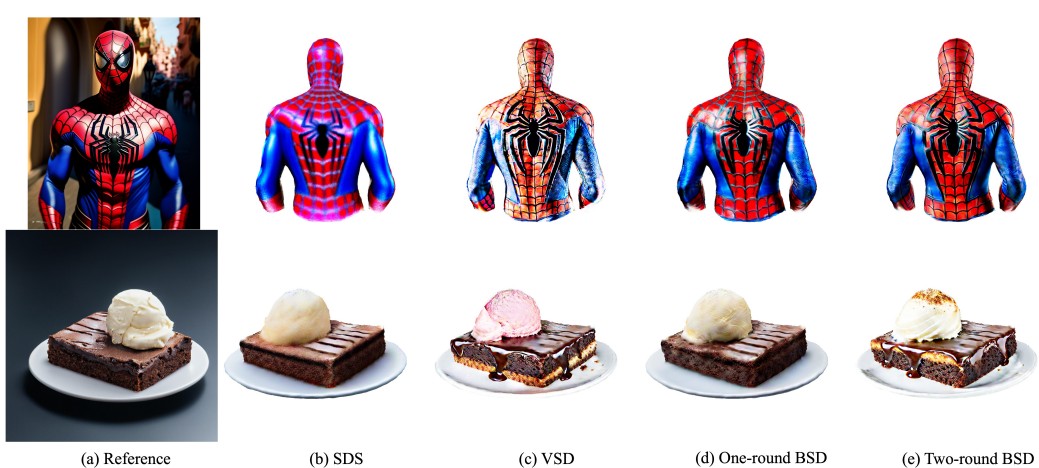

| (a) Reference | (b) SDS | (c) VSD | (d) One-round BSD | (e) Two-round BSD |

Figure 10: Ablation study of the effectiveness of the proposed BSD (Bootstrapped Score Distillation). From left to right are: (a) reference image. (b) texture optimization with SDS loss. (c) The use of VSD loss results in richer texture detail but comes with texture inconsistency. (d) Application of BSD improves texture consistency with one round of DreamBooth. (e) Applying two rounds of BSD adds more detail to the final result.

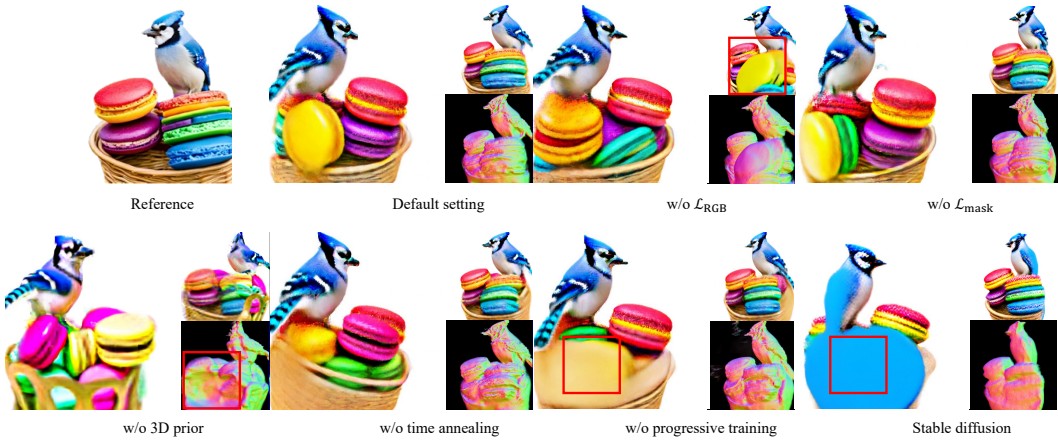

| Reference | Default setting | w/o $\mathcal{L}_{RGB}$ | w/o $\mathcal{L}_{mask}$ |

| w/o 3D prior | w/o time annealing | w/o progressive training | Stable diffusion |

Figure 11: Ablation studies of design choices in the geometry sculpting stage.

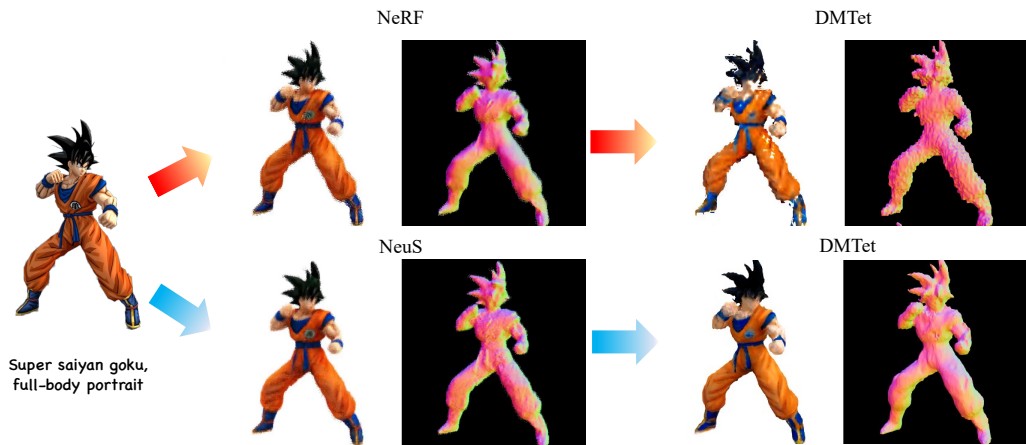

Figure 12: Ablation studies of the representation choice. Compared with NeRF, NeuS models clear surfaces and enables extracting higher-quality surfaces for DMTet initialization.

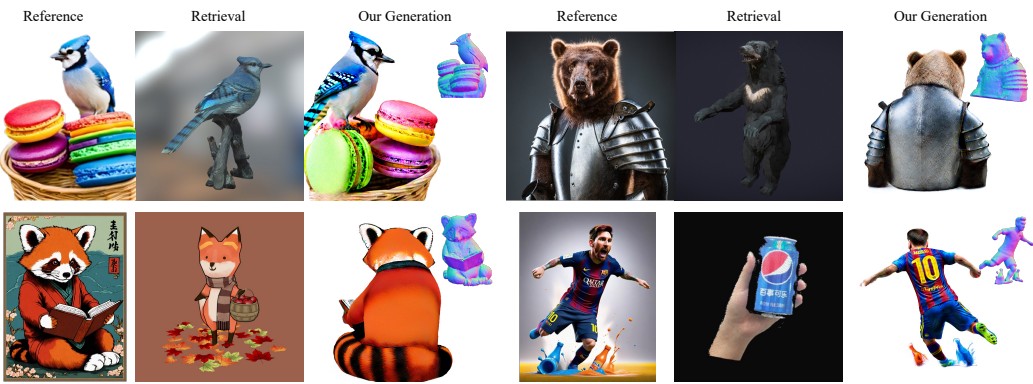

Figure 13: Comparison between the object retrievals within Objaverse by the test bench images and our generations. Our method is robust to the out-of-domain reference images.

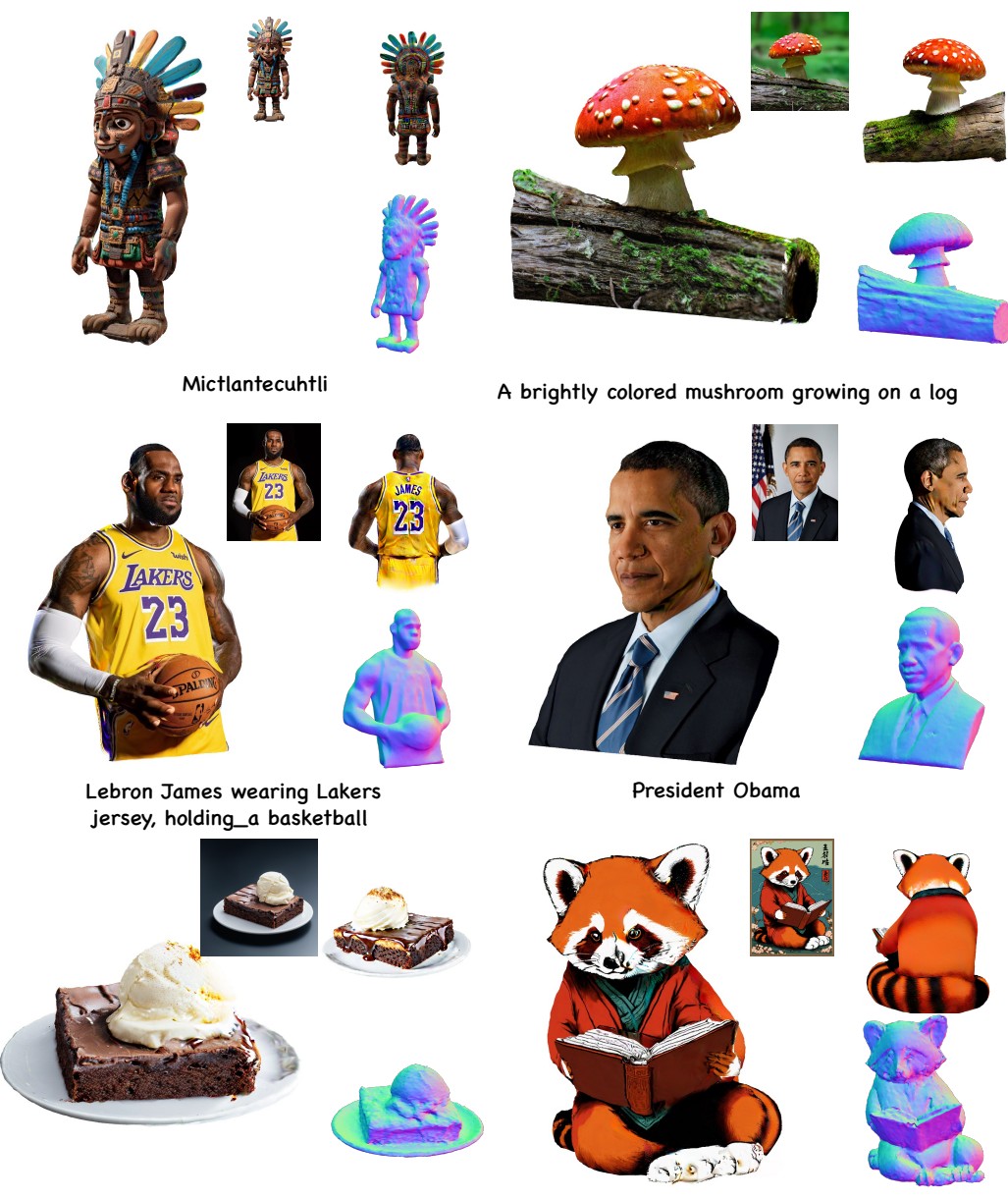

Mictlantecuhtli

A brightly colored mushroom growing on a log

Lebron James wearing Lakers jersey, holding_a basketball

President Obama

A DSLR photo of a delicious chocolate brownie dessert with ice cream on the side

Wes Anderson style Red Panda, reading a book, super cute, highly detailed and colored

Figure 14: Additional results of DreamCraft3D.

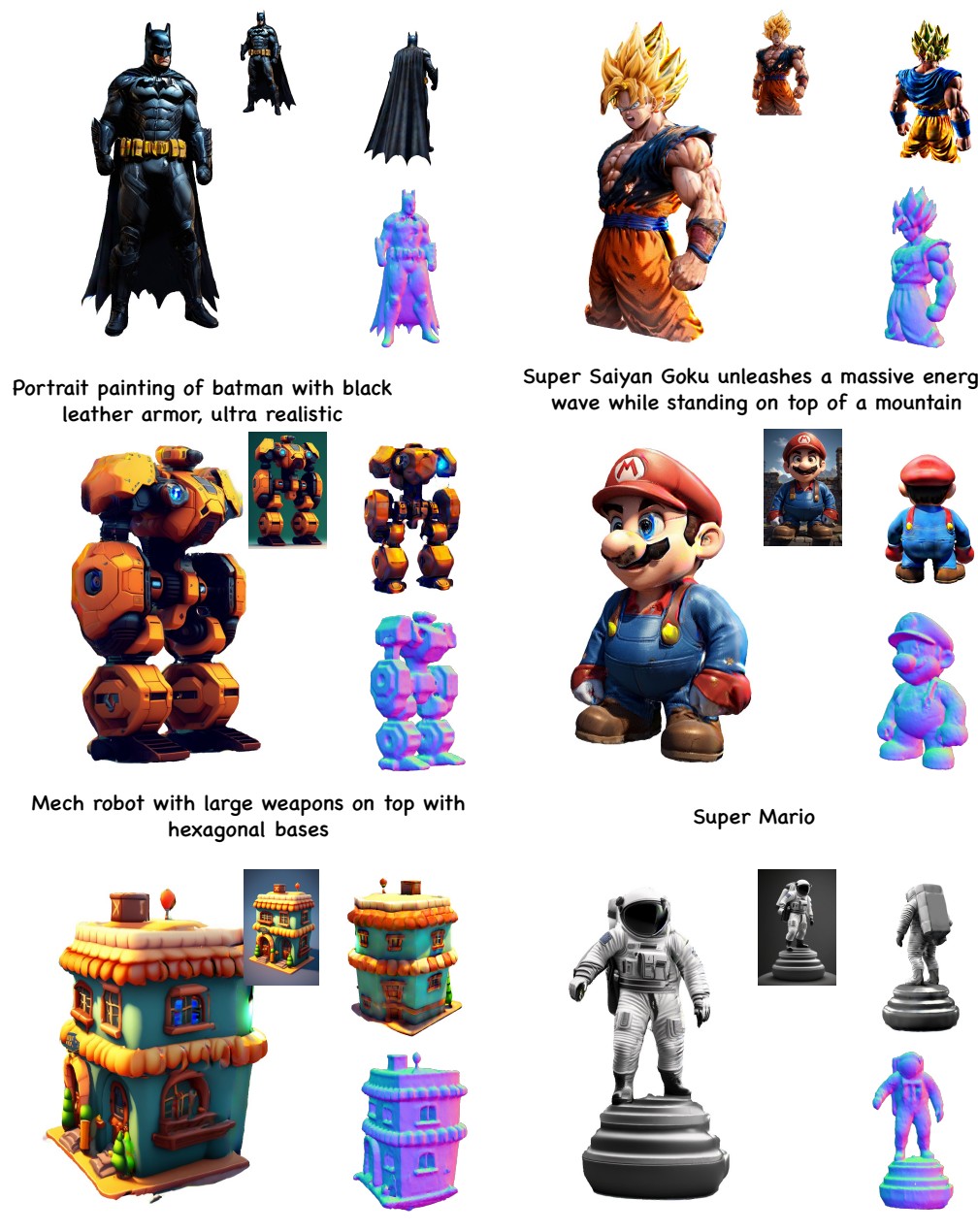

Figure 15: Additional results of DreamCraft3D.

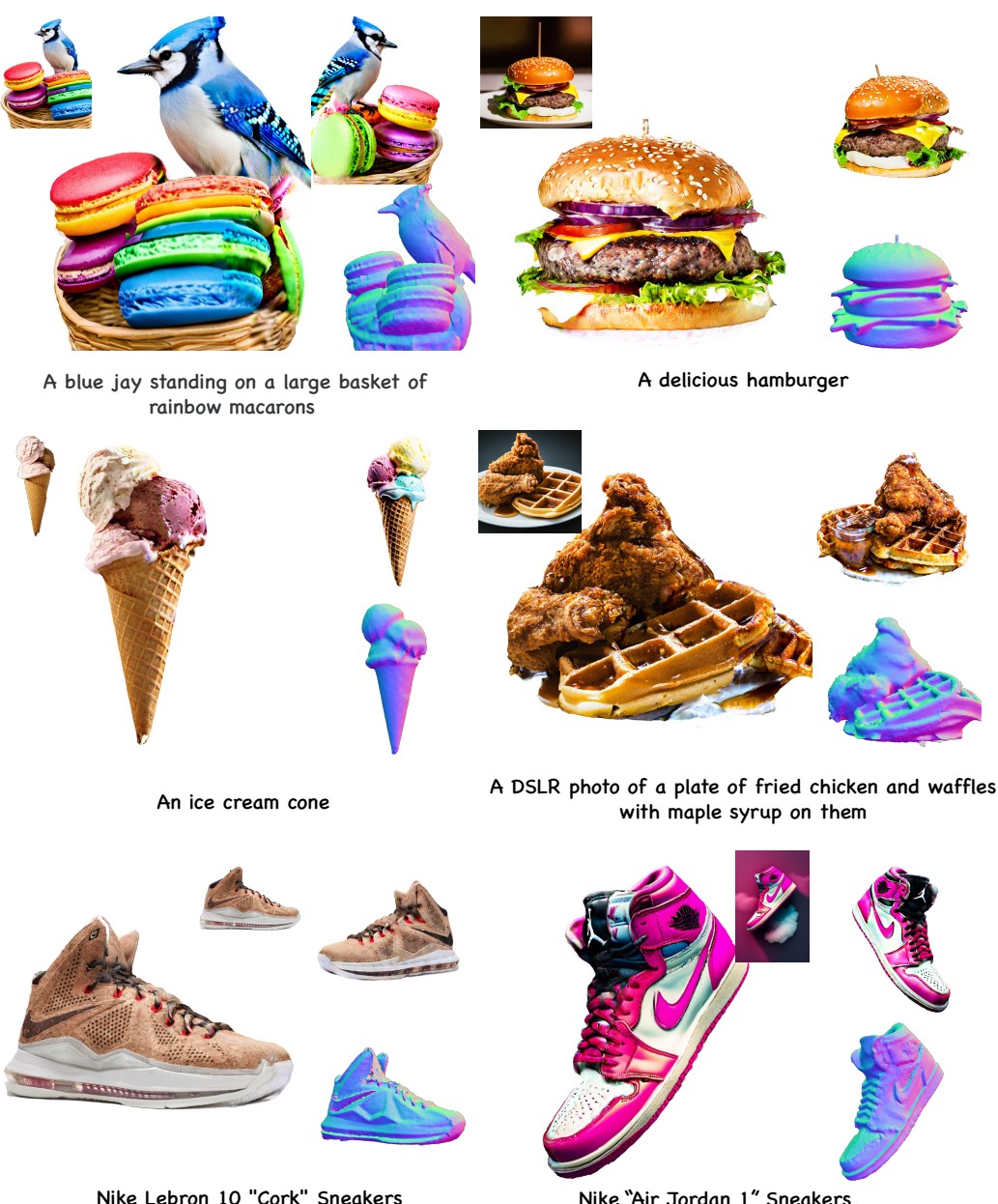

A blue jay standing on a large basket of rainbow macarons

A delicious hamburger

An ice cream cone

A DSLR photo of a plate of fried chicken and waffles with maple syrup on them

Nike Lebron 10 "Cork" Sneakers

Nike "Air Jordan 1" Sneakers

Figure 16: Additional results of DreamCraft3D.

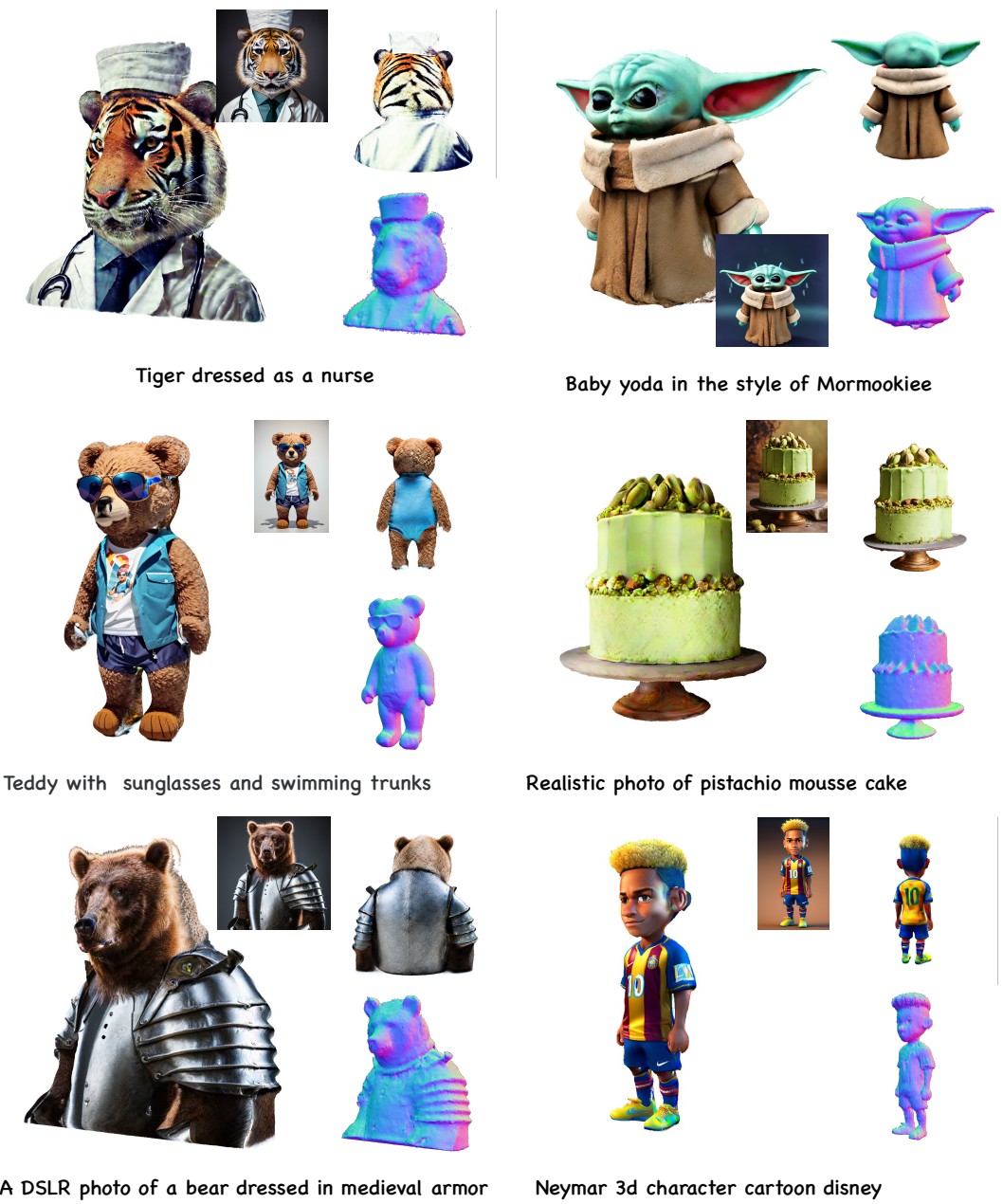

Figure 17: Additional results of DreamCraft3D.

