# OpenReview forum: "DreamCraft3D: Hierarchical 3D Generation with Bootstrapped Diffusion Prior"
_ICLR.cc/2024/Conference — ICLR 2024 poster_

### Official Review · Reviewer_NmnF · 2023-10-27

**Soundness:** 3 good
**Presentation:** 3 good
**Contribution:** 2 fair
**Rating:** 6
**Confidence:** 3

**Summary:**

This paper proposes a hierarchical way for 3D content generation from text inputs, following the line of dreamfusion.
Instead of using 2D diffusion only, it combines zero123 to give 3D priors.
It also proposes a bootstrapped score sampling method, which finetunes the stable diffusion using dreamboth during the distillation  process.
Combining carefully-tuned parameters, this paper achieves impressive results of text-to-3D.

**Strengths:**

1. The results of this paper is impressive. The improvement seems significant compared to previous methods.
2. This paper combines many tricks in a convincing way, and clearly shows the effectiveness of each trick.
3. The paper is well written.

**Weaknesses:**

1. The running time of methods. This method involves several stages an looks quite complicated. I am curious about how long does it take to finish the whole process? And how does it compare to other baselines?
2. It would be more clear if the difference between the proposed bootstrapped diffusion prior and LORA updates in ProlificDreamer is elaborated, since they both update the diffusion during the distillation.  What's intuitive difference and actual different in implementation?
3. I understand it is hard to evaluate generation task, but more details about metrics used in Table 1 would be convincing. All CLIP, Contextual ,PSNR and LPIPS are image-level evaluation, and it is not clear how to convert them to 3D level. PSNR and LPIPS are measured on reference images, which is understandable. But how is the CLIP and Contextual calculated? How many views are rendered from each scene for evaluation?

**Questions:**

See weakness.

---

> ### Author Response · Authors · 2023-11-21
> **Response to Reviewer NmnF**
>
> Thanks for your effort invested in reviewing our work. We are grateful for your recognition of the novelty and advancements our research brings to the field.  In response to your questions, please find our clarifications below.
>
>
> **Q1: Runtime of the method.**
>
> A: The training time comparison against recently published works is shown below.
>
>
> | Method          | Training Time |
> | --------------- | ------------- |
> | DreamCraft3D    | 2.3h          |
> | Magic123        | 2.8h          |
> | Magic3D         | 1.5h          |
> | ProlificDreamer | 10h           |
> | DreamFusion     | 0.7h          |
>
> Below we report the detailed runtime profile. These numbers are measured on 30 objects. The training is conducted on 8 A100 GPUs and the inference on 512 $\times$ 512 resolution is measured using a single A100 GPU. We did not fully optimize the training time, and we expect significant speedup with further engineering optimization.
>
> | Running Type                       | Time  |
> | ---------------------------------- | ----- |
> | Training Time (total)              | 2.3h  |
> | Training Time (Geometry Sculpting) | 1.5h  |
> | Training Time (Texture)            | 0.8h  |
> | Inference Time                     | 0.05s |
>
> **Q2: The difference between the proposed bootstrapped diffusion prior and the LoRA updates in ProlificDreamer.**
>
> A: Thank you for raising the point. Below we clarify their difference.
>
> ProlificDreamer regards the 3D solution corresponding to the text as a distribution. Thus, ProlificDreamer aims to learn a distribution $\mu$ to match the distribution characterized by the pretrained diffusion model. Mathematically, this is achieved by minimizing the score function of the two distributions. The score function of the pretrained model is represented by its noise prediction $\epsilon_{\text{pretrain}}(x_t,t,y)$. On the other hand, the score function of the optimized 3D scene is estimated by the LoRA model, *i.e.*,$\epsilon_\\phi\left(x_t, t, c, y, \\{x\\}\right)$, using the multi-view image renderings $\\{x\\}$. Hence, the VSD loss used in ProlificDreamer is:
> $$
> \nabla_\theta \mathcal{L}_{\mathrm{VSD}}(\theta) \triangleq \mathbb{E}\_{t, \epsilon, c}\left[\omega(t)(\epsilon\_{\mathrm{pretrain}}\left(x_t; t, y\right)-\underbrace{\epsilon\_\phi\left(x_t; t, c, y, \\{x\\}\right)}\_{\text {LoRA }}) \frac{\partial g(\theta, c)}{\partial \theta}\right].
> $$
>
> The proposed bootstrapped score distillation differs in the motivation fundamentally. We claim that a fixed 2D diffusion prior is not 3D aware and cannot provide sufficient guidance that ensures multi-view consistency. Instead, we propose to learn from a moving target that also updates based on the 3D scene being optimized. In the implementation, we render the multi-view images of the scene and augment their quality (*i.e.*, adding the Gaussian noise and then denoising them) using a diffusion model, forming high-quality augmented renderings denoted as $\\{\tilde{x}\\}$. These augmented images are used to compute the LoRA model that serves as the training target (the first term in the following formula):
> $$
> \nabla_\theta \mathcal{L}_{\mathrm{BSD}}(\theta) \triangleq \mathbb{E}\_{t, \epsilon, c}\left[\omega(t)(\underbrace{\epsilon\_{\mathrm{DreamBooth}}\left(x_t; t, c, y,  \\{\tilde{x}\\}_k\right)}\_{\text{BSD}}-\underbrace{\epsilon\_\phi\left(x_t; t, c, y, \\{x\\}\right)}\_{\text {LoRA }}) \frac{\partial g(\theta, c)}{\partial \theta}\right]
> $$
> We iterate the above process and apply attenuated augmentation strength to derive augmented views, $\\{\tilde{x}\\}_k$, where $k$ denotes the iteration steps. As training proceeds, both the diffusion prior and the 3D scene get improved, which bootstraps the 3D optimization. We have better formulated the above process during revision.
>
>
> **Q3: Further explanation of the evaluation protocol.**
>
> A: Thanks for the question. We compute the CLIP score to evaluate the semantic similarity between the image renderings at novel views and the reference image. A higher CLIP score indicates better semantic consistency around different views. The contextual loss is computed to measure the style similarity between the novel views and the reference image. The contextual metric can be regarded as an improved LPIPS without requiring pixel-wise alignment. A higher contextual metric means improved style consistency around the generated 3D asset. We use 120 rendered views to compute both the CLIP and the contextual score. To better reveal the subjective quality, we also conducted a user study which gathered 480 responses from a total of 32 participants. We have included these details in the paper revision.

---

> > ### Comment · Reviewer_NmnF · 2023-12-05
> > **Reply to authors' reply**
> >
> > Thanks for the clarification. After reading all the other reviews and authors' replies, I am positive about this paper and maintain the current score. Although I am concerned with the novelty and (might be over-complicated) components combination of this paper, this paper indeed achieves very impressive results. I hope authors could give more clear explanations and ablations about the method if it is accepted. Thanks

---

### Official Review · Reviewer_GYis · 2023-11-01

**Soundness:** 3 good
**Presentation:** 3 good
**Contribution:** 1 poor
**Rating:** 5
**Confidence:** 3

**Summary:**

The paper proposes an image-conditioned 3D generative model and focuses on improving view consistency upon current techniques. It proposes to use a view-conditioned 3D prior, and various techniques to improve the texture fidelity. Results show empirical benefits of the proposed method compared to prior arts.

**Strengths:**

* Qualitative results selected in the paper show a large margin of advantages of the proposed method compared to the baselines.
* The paper focuses on addressing the view consistency problem, and the examples shown in the paper provide strong evidence for this claim.

**Weaknesses:**

* The paper combines several existing techniques, including Zero-1-to-3, DreamBooth, VSD, and a bag of tricks including progressive view sampling and a transition between different 3D representations. The novelty of this paper is limited.
* The pipeline depends on Zero-1-to-3 to provide a good initial shape. For scenes that Zero-1-to-3 fails on, e.g. scenes that differ greatly from the synthetic training distribution of Zero-1-to-3, it's unclear if the proposed method can still generate reasonable results.
* Multiple loss or gradient terms are introduced in Section 4.1, but the relative weights of these terms are not described.

**Questions:**

* Do all examples shown in the paper share the same hyperparameter configuration?

---

> ### Author Response · Authors · 2023-11-21
> **Response to Reviewer GYis**
>
> We are grateful for your recognition to our paper results and the effectiveness of our approach. Please find our point-to-point response to the review comments.
>
> **Q1: Technical contribution.**
>
> A: Thanks for your question. 3D generation is particularly challenging due to the lack of a large quantity of 3D data. This paper aims to present two key components that lead to high-fidelity 3D content creation.
>
> First, **we demonstrate how a meticulously designed hierarchical framework can achieve high-fidelity 3D generation**. We decompose the whole generation process into multiple steps: 2D image creation from the text prompt, the geometry sculpting stage and texture boosting stage, with each stage having different priorities. The geometry sculpting stage focuses on obtaining a coherent geometry. we show that a  3D-aware diffusion prior, accompanied with multiple effective techniques like progressive view and diffusion timestep annealing, can lead to consistent 3D geometry. While this stage often compromises the texture, we resort to the following texture boosting stage which is dedicated to improving the texture quality. We believe this hierarchical framework with thoughtful design choices sets up a good starting point for the following research works.
>
> Second, we propose a bootstrapped score distillation (BSD) scheme that considerably enhances the texture quality. **The key motivation is that a fixed 2D diffusion model does not suffice to provide view-consistent guidance. We propose to improve this diffusion prior and the 3D mesh in the cyclic optimization: steadily improved multiview renderings lead to an increasingly view-consistent diffusion prior, which in turn further improves the 3D texture. In this way, the 3D optimization is bootstrapped, ultimately leading to highly detailed textures.** In the implementation, we render the multi-view images of the scene and augment their quality (*i.e.*, adding the Gaussian noise and then denoising them) using a diffusion model, forming high-quality augmented renderings denoted as {$\{\tilde{x}\}$}. These augmented images are used to compute the LoRA model that serves as the training target:
> $$\nabla\_\theta \mathcal{L}\_{\mathrm{BSD}}(\theta) \triangleq \mathbb{E}\_{t, \epsilon, c}\left[\omega(t)(\underbrace{\epsilon\_{\mathrm{DreamBooth}}\left(x_t; t, c, y,  \\{\tilde{x}\\}_k\right)}\_{\text{BSD}}-\underbrace{\epsilon\_\phi\left(x_t; t, c, y, \\{x\\}\right)}\_{\text {LoRA }}) \frac{\partial g(\theta, c)}{\partial \theta}\right]$$
>
> **This technique substantially differs from score distillation methods in that it optimizes against a continuously evolving target.** This technique is vital to compelling visual quality and we ablate its effectiveness through Figure 5, Figure 6, Figure 10 and Table 2. To the best of our knowledge, this method has not been explored before.
>
>
> Hence, we assert this paper is not a trivial combination of previous works. Rather, we believe it represents a significant contribution to the field through the introduction of the aforementioned key components. We hope that these insights will be valuable to the community.
>
>
> **Q2: Performance of out-of-domain generation.**
>
> A: That's a good point! As illustrated in the paper, our method demonstrates good imaginative and combinational power and can produce fantasy 3D assets. We have provided more qualitative results in the appendix. We welcome the reviewer to check them.
>
> To better explain that our results greatly differ from the existing 3D data, we retrieve the top-1 similar image from the Objectverse dataset. The retrieval results are shown in Figure 13 in the appendix. This retrieval experiment shows that our results are far different from the instance in the 3D dataset, showing good out-of-domain generalization capability for our approach.
>
>
> **Q3: Relative weights of loss terms.**
>
> A: Thanks for figuring it out. We have included this implementation detail in Section 4.2 during revision.
>
> **Q4: Do all the examples share the same parameters?**
>
> A: Thanks for raising the point. Most cases in our paper (roughly 90%) share the same hyper-parameters. However, some adjustments might be needed. When the reference is captured at an oblique angle, we may change the evaluation angle for the view-conditioned diffusion model. Additionally, we might need to properly set the camera FOV to suit the object size in the reference image.

---

### Official Review · Reviewer_vdnk · 2023-11-01

**Soundness:** 3 good
**Presentation:** 3 good
**Contribution:** 3 good
**Rating:** 6
**Confidence:** 5

**Summary:**

The authors present a framework for text-to-3D generation. This is achieved by first synthesizing a single view image from a text point and using a view conditioned diffusion model to guide the geometry generation process. In particular, the framework uses a combination of SDS losses from a view-conditioned diffusion model and a text conditioned diffusion model. Further, the renderings from the 3D representation are used as additional training data to train a personalization model to improve texture quality. State of the art results are demonstrated on text-to-3D creations compared to several recent baselines.

**Strengths:**

1. **Result Quality** : The quality of the generated assets are very impressive. Particularly, in examples where the back side of the object hallucinates necessary details.
2. **Novelty** : The proposed approach is reasonably novel as it proposes a combination of 2D SDS and 3D aware SDS losses for better geometric reconstruction. The paper also compares against contemporary work with similar ideas (Magic123) and shows state of the art results against the same.
2. **Related work**: An adequate treatment of the related work has been provided. Efforts have been made to reference and compare against contemporary unpublished work that also have compelling results.
3. **Reproducibility**: All the implementation is based off of available open source code bases, make it easy to reproduce. Network and training details have been provided to further aid in reproducibility.
4. **Progressive view training**: Is a simple and elegant idea to make sure that the generated geometry is not flat.
5. **Benchmark dataset**: This work shares a evaluation benchmark of 300 prompt image pairs, which can be used to evaluate future work in text-to-3D synthesis.
6. **Structure aware latent regularization**: Is another interesting strategy for making sure already generated texture information is preserved. This section would benefit from shifting it from the appendix to the main section.

**Weaknesses:**

1. **Multi component-Combination** : Although reasonably novel, the framework uses several different diffusion networks for different components. Particularly, Deep-IF for initial image and base 3D geometry generation, SD for personalization and Zero-123 for viewpoint guided SDS. The final framework appears to be a combination of Magic123, Zero1-to-3, ProlificDreamer and Fantasia3D[1]
1. **Claims** : The authors claim that personalizing the diffusion model based on the multiview renderings helps improve texture, however, this seems counter intuitive, since the initial texture generated from the 3D representation are expected to be worse than the high quality texture. Providing additional insights about this would be helpful.
2. **Writing**: The manuscript contains certain syntactic and language errors (such as some of the nits indicated below) and would benefit from a thorough proof reading pass of both the main paper and the appendix.
3. **Ablations**: Qualitative ablations are provided for some of the components. But adding some quantitative ablations that show change in quality would be helpful. Particularly, metrics for 3D consistency for number of BSD steps, texture quality with and without BSD and effect of progressive training, timestep annealing and choice of diffusion model used for SDS (SD vs DeepFloyd).
4. **Training and inference time costs**: A comparison of training and inference time cost and memory footprint of the proposed approach compared to the baselines is also important and would provide some insights about the inference time trade off and memory required to achieve this quality.


Nits:
Sec 4. "in the next..." -> "in the next section?" or "next".
Sec 4.2 "multiview rendering from last stage.." -> "from the previous stage?"


[1] Chen et al. Fantasia3D, ICCV23

**Questions:**

What is the effect of time step annealing? and progressive viewpoint training?

---

> ### Author Response · Authors · 2023-11-21
> **Response to Reviewer vdnk**
>
> We are grateful for your acknowledgment of the technical novelty and results presented in our work. Please find our response to your questions as follows.
>
>
> **Q1: The use of multiple diffusion models.**
>
> A: Thanks for your comment. This paper presents two key components for high-fidelity 3D generation: hierarchical generation and bootstrapped score distillation (BSD).
>
> Our hierarchical approach involves decomposing the challenging 3D generation into multiple well-designed stages. The coarse geometry sculpting stage aims to learn a coherent geometry, so we introduce a 3D-aware diffusion prior, along with several effective techniques such as progressive viewing and diffusion timestep annealing, to ensure the creation of a coherent geometry. The subsequent stage further refines the texture, demonstrating how a meticulously designed hierarchical framework can generate high-fidelity 3D assets of unprecedented quality.
>
> Second, we propose a novel bootstrapped score distillation (BSD), which fundamentally differs from prior SDS-based methods in that we optimize against a dynamically evolving target. This bootstrapped diffusion prior substantially improves the realism of the 3D texture. To the best of our knowledge, this technique has not been explored before.
>
> Therefore, we assert that our method is not merely a trivial combination of existing approaches. Rather, this work builds on the merits of prior works while presenting novel insights, setting a new benchmark in the field.
>
>
> **Q2: How does the personalized diffusion model trained on initial texture boost the texture quality?**
>
> A: Thanks for raising the point. Indeed, the texture from the previous stage has limited quality and suffers from blurriness. Therefore, **we use a pretrained diffusion model to enhance their quality**.
>
> We incorporate Gaussian noise into multi-view images, allowing a diffusion model to reconstruct high-quality visuals. Despite affecting view consistency, this process yields impressively realistic images. The enhanced multi-view images introduce scene aspects to form a personalized diffusion model, which significantly improves texture quality.
>
> We iterate the above process several times and each step leads to improved 3D texture. We progressively decrease the strength to augment the multi-view rendering to better retain their view consistency. As we iterate the above process, we bootstrap the texture quality for the 3D instance being optimized.
>
>
> **Q3: Writing.**
>
> A: Thanks for your comment. We have conducted careful proofreading and fixed some typos in the revision.
>
> **Q4: More ablation studies.**
>
> A: Per your suggestion, we provide additional qualitative and quantitative ablation studies for two stages respectively as follows. The techniques of time annealing and progressive training improve the view consistency relative to the reference image, as measured by the CLIP and contextual score. We have included these results in the appendix.
>
> |               | CLIP$\uparrow$  | Contextual $\downarrow$ | PSNR $\uparrow$  | LPIPS $\downarrow$ |
> | ------------- | ----- | ---------- | ------ | ------ |
> | Default setting | **0.745** | **1.936**  | **20.836** | 0.031  |
> | w/o $\mathcal{L}_{\textnormal{RGB}}$ | 0.619 | 2.212 | 10.562 | 0.095  |
> | w/o $\mathcal{L}_{\textnormal{mask}}$| 0.736 | 1.949 | 19.307 | 0.057  |
> | w/o 3D prior  | 0.679 | 2.105 | 20.252 | 0.092 |
> | w/o time annealing | 0.737 | 1.943 | 20.120 | 0.035 |
> | w/o progressive training | 0.721 | 1.962 | 20.317 | **0.029** |
> | Stable diffusion | 0.669 | 2.154      | 18.182 | 0.099  |
>
> |               | CLIP$\uparrow$  | Contextual $\downarrow$ | PSNR $\uparrow$  | LPIPS $\downarrow$ |
> | ------------- | ----- | ---------- | ------ | ------ |
> | SDS           | 0.775 | 1.894      | 25.258 | 0.036  |
> | VSD           | 0.792 | 1.833      | 31.426 | 0.005  |
> | One-round BSD | 0.863 | 1.647      | 30.431 | 0.008  |
> | Two-round BSD | **0.896** | **1.579**      | **31.801** | **0.005**  |
>
>
> **Q5: Training and inference time costs.**
>
> A: The training time comparison against recently published works is shown below.
>
>
> | Method          | Training Time |
> | --------------- | ------------- |
> | DreamCraft3D    | 2.3h          |
> | Magic123        | 2.8h          |
> | Magic3D         | 1.5h          |
> | ProlificDreamer | 10h           |
> | DreamFusion     | 0.7h          |
>
> We report the detailed runtime profile below. These numbers are measured on 30 objects. The training is conducted on 8 A100 GPUs and the inference on 512 $\times$ 512 resolution is measured using a single A100 GPU. We did not fully optimize the training time and we expect significant speedup with further engineering optimization.
>
> | Running Type                       | Time  |
> | ---------------------------------- | ----- |
> | Training Time (total)              | 2.3h  |
> | Training Time (Geometry Sculpting) | 1.5h  |
> | Training Time (Texture)            | 0.8h  |
> | Inference Time                     | 0.05s |

---

> > ### Comment · Reviewer_vdnk · 2023-11-23
> > **Response to questions**
> >
> > The authors do a good job of addressing all the questions raised. To that end I will retain the current score and recommend the work for acceptance.

---

### Official Review · Reviewer_c9Ad · 2023-11-05

**Soundness:** 3 good
**Presentation:** 3 good
**Contribution:** 3 good
**Rating:** 8
**Confidence:** 4

**Summary:**

This paper studies the task of text-to-3D generation, incorporating a series of enhancements that yield highly promising results. The introduced framework DreamCraft3D generates 3D assets with remarkable fidelity and minimal artifacts, marking a great advancement in the field. Key innovations of DreamCraft3D encompass initial reference image generation, a geometry sculpting stage, and a subsequent texture boosting stage.

Through an extensive series of experiments, DreamCraft3D emerges as a formidable contender, outperforming both optimization-based methods such as DreamFusion and single-image-based 3D reconstruction techniques exemplified by Magic123.

----- After rebuttal ------
Thanks for providing more experiments and most of my concerns/doubts are addressed. I have raised my rating to 8-good paper.  I suggest the authors to include the extra ablation studies into the main paper, which can be informative to many readers.

**Strengths:**

This paper demonstrates several notable strengths, as outlined below:

1. **Impressive Qualitative Results**: Unlike existing methods, the generated 3D assets in this study exhibit great fidelity and significantly fewer artifacts. Furthermore, the Janus problem is substantially mitigated, marking a substantial improvement.

2. **Comprehensive Comparative Analysis**: The paper conducts comparisons with both optimization-based methods and single-image-based 3D reconstruction techniques, enhancing the credibility of the experimental findings.

3. **Effective and Reasonable Solutions**: The proposed solutions and submodules are not only reasonable in design but also demonstrate practical effectiveness.

4. **Quantitative Evaluation and Benchmarking**: Despite the inherent challenges in evaluating text-to-3D methods, this paper makes a commendable effort to perform quantitative comparisons and establishes a new benchmark, contributing to the field's progress.

5. **Clarity in Communication**: The writing in this paper is mostly clear and easy to follow.

**Weaknesses:**

Even though the final results are encouraging, there are some weaknesses that need to be addressed:

1. **Clarity of Approach Section**: The introduction of multiple stages and new sub-modules (loss functions, optimization stages) makes the presentation in the approach section less clear. It would be beneficial to include an overview section with a clear definition of the total loss and a pseudo-algorithm summarizing the steps in different stages.

2. **Technical Contribution Clarification**: The major technical contributions are not clearly defined. It's essential to clarify which submodules or loss functions represent the most significant changes and contribute the most to the final performance. This information is crucial for a deeper understanding of the work. Overall, the proposed framework is a bit complicated and contains many modules and stages.

3. **Incomplete Validation**: Some modules are not thoroughly validated in the experiments. For instance, the importance and effects of the two losses in Eq.4, the L_RGB, and L_mask losses should be elaborated upon. It's also important to explain the advantages of using NeuS compared to NeRF-variants and provide validation results. Additionally, details on the implementation of "Progressive view training" should be provided.

4. **Prompt Engineering Clarity**: The paper needs to address discrepancies in prompt engineering. As shown in Fig.2, the prompt for generating the reference image is "an astronaut in a sand beach," while for dreambooth, it's "an [v] astronaut." These differences need clarification to ensure consistency and reproducibility.

**Miscellaneous:**

- **Figure Order**: The order of figures is confusing. Fig.4 is mentioned later in the text than Fig.5/6 but appears earlier. Consider reordering them for coherence.

- **Fig.6 Text Descriptions**: Fig.6 is missing text descriptions of the corresponding stages. It's unclear which figure corresponds to what stage, making it difficult to follow.

- **Fig.5 Ablation Study**: The ablation study in Fig.5 could be expanded. Since text-to-3D is challenging to evaluate, a single example may not provide enough information. More comprehensive insights can help to strengthen the research.

**Questions:**

1. **training/inference time**: what's the training time of the entire pipeline and what's the time for each individual stage? Meanwhile, how
1. **Training and Inference Time**: Could you provide insights into the training time for the entire pipeline and the time required for each individual stage? Additionally, how long does it take to generate a new 3D model from scratch, and could you provide a breakdown of the time allocation for this process?

2. **Robustness of the System**: Given the numerous modules and training stages, how robust is the system? Are the same set of parameters applicable to different methods, and if not, what are the characteristics of failure cases? Have you observed any convergence issues in any of the stages?

3. **Choice of Deepfloyd IF**: What is the rationale for not using SD in the first stage and opting for Deepfloyd IF instead? Could you elaborate on the advantages of Deepfloyd IF in this context?

---

> ### Author Response · Authors · 2023-11-21
> **Response to Reviewer c9Ad (Part 1/2)**
>
> Thank you for the kind words for recognizing the impact of the paper. Please find our point-to-point response to your questions below.
>
>
> **Q1: Clarity of approach section.**
>
> A: Thanks for the comment. As suggested, we summarize the loss functions in the two stages in Eq.7 and Eq.9, respectively. The algorithm for the bootstrapped score distillation is detailed in the appendix. Furthermore, we have supplemented more implementation details. We hope these changes have improved the clarity of our method.
>
>
> **Q2: Technical contribution clarification.**
>
> A: Our method addresses 3D generation through a two-tiered process: a geometry sculpting stage and a texture boosting stage.
>
> In the geometry sculpting stage, we strive to construct a coarse but 3D-coherent geometry. This is achieved through a 3D-aware diffusion prior, augmented by several effective training strategies. This stage aims to create robust and coherent geometry.
>
> For the texture boosting stage, we introduce a novel bootstrapped score distillation loss. This approach enables the generation of detailed and visually stunning textures, contributing to a more realistic and high-quality 3D output.
>
> Hence, the two stages focus on different aspects: the former prioritizes geometry while the latter adds meticulous texture. To explain the relative importance of different techniques, we conduct a more thorough ablation study, both qualitatively and quantitatively, in Figure 11, Table 2 and Table 3, as shown in the appendix. For your convenience, we provide quantitative results for the BSD loss as follows:
>
> |                                       | CLIP$\uparrow$ | Contextual $\downarrow$ | PSNR $\uparrow$ | LPIPS $\downarrow$ |
> | ------------------------------------- | -------------- | ----------------------- | --------------- | ------------------ |
> | Default setting                       | **0.745**      | **1.936**               | **20.836**      | 0.031              |
> | w/o $\mathcal{L}_{\textnormal{RGB}}$  | 0.619          | 2.212                   | 10.562          | 0.095              |
> | w/o $\mathcal{L}_{\textnormal{mask}}$ | 0.736          | 1.949                   | 19.307          | 0.057              |
> | w/o 3D prior                          | 0.679          | 2.105                   | 20.252          | 0.092              |
> | w/o progressive training              | 0.721          | 1.962                   | 20.317          | **0.029**          |
> | w/o time annealing                    | 0.737          | 1.943                   | 20.120          | 0.035              |
> | Stable diffusion                      | 0.669          | 2.154                   | 18.182          | 0.099              |
>
> |               | CLIP$\uparrow$  | Contextual $\downarrow$ | PSNR $\uparrow$  | LPIPS $\downarrow$ |
> | ------------- | ----- | ---------- | ------ | ------ |
> | SDS           | 0.775 | 1.894      | 25.258 | 0.036  |
> | VSD           | 0.792 | 1.833      | 31.426 | 0.005  |
> | One-round BSD | 0.863 | 1.647      | 30.431 | 0.008  |
> | Two-round BSD | **0.896** | **1.579**      | **31.801** | **0.005**  |
>
> **Q3: Further validation, especially for $L_{\textnormal{RGB}}$, $L_{\textnormal{mask}}$ and the use of NeuS representation.**
>
> A: We add both quantitative and qualitative ablation studies of loss functions ($\mathcal{L}\_{\textnormal{RGB}}$ and $\mathcal{L}_{\textnormal{mask}}$), training strategies (timestep annealing and progressive training), and model choice (DeepFloyd vs. Stable Diffusion) in the revision (See Appendix A.4), showing the effectiveness of individual techniques.
>
> We choose to optimize NeuS instead of NeRF in the geometry sculpting stage because NeRF learns a volume field which is hard to yield a high-quality surface for the following stage, whereas NeuS parameterizes the scene with neural SDF representation, yielding a high-quality surface for the following stage. Please refer to Figure 12 in the Appendix for visual examination.
>
> For progressive training, we linearly increase the sampling range of camera positions with elevation angle ($\phi_\textnormal{cam}$) from $0^{\circ}$ to  $[-10^{\circ}, 45^{\circ}]$, and azimuth angle ($\theta_\textnormal{cam}$) from $0^{\circ}$ to $[-180^{\circ}, 180^{\circ}]$. The progress viewing training takes 200 iterations. We have included this implementation detail in the Appendix following your suggestion.

---

> ### Author Response · Authors · 2023-11-21
> **Response to Reviewer c9Ad (Part 2/2)**
>
> **Q4: Prompt engineering clarity.**
>
> A: We are thankful for your careful examination. We use the prompt "a [V] astronaut" rather than "an astronaut in a sand beach" because we focus on hallucinating the 3D for the foreground object exclusively. The unique identifier “[v]” is used to characterize subject-specific attributes such as lighting, color, pose, etc.
>
>
> **Q5: Training and inference time.**
>
> A: We compare the training time  against recent leading methods:
>
>
> | Method          | Training Time |
> | --------------- | ------------- |
> | DreamCraft3D    | 2.3h          |
> | Magic123        | 2.8h          |
> | Magic3D         | 1.5h          |
> | ProlificDreamer | 10h           |
> | DreamFusion     | 0.7h          |
>
> The profile of training and inference time is reported in the following table. We report the average runtime on 30 objects. The 3D asset is optimized on 8 A100 GPUs and the inference is done using a single GPU. We render at 512 $\times$ 512 resolution during inference. We haven't carefully optimized the optimization speed we consider efficient 3D generation as future work.
>
> | Running Type                       | Time  |
> | ---------------------------------- | ----- |
> | Training Time (total)              | 2.3h  |
> | Training Time (geometry sculpting) | 1.5h  |
> | Training Time (texture boosting)            | 0.8h  |
> | Inference Time                     | 0.05s |
>
>
> **Q6: Robustness of the system.**
>
> A: This is a good point! Most cases in our paper (roughly 90%) share the same hyper-parameters. However, some adjustments might be needed. When the reference is captured from an oblique angle, we may change the evaluation angle for the view-conditioned diffusion model. Additionally, we might need to properly set the camera FOV to suit the object size in the reference image.
>
>
>
> **Q7: The rationale of using Deepfloyd IF.**
>
> A: We employ the DeepFloyd IF model as it provides enhanced view-awareness and reduces the likelihood of encountering the “Janus (multi-face) problem”. This is because the DeepFloyd IF model has a stronger language encoder (T5-XXL) that better encodes the pose information denoted in the text prompt. Moreover, the DeepFloyd base model offers gradient on a coarse resolution (64x64), which promotes a coherent 3D geometry rather than introducing high-frequency signals that distract the geometry sculpting. The ablation of this design choice is shown in Table 3 in the revised version.
>
> **Q8: Miscellaneous.**
>
> A: We have improved the order of figures and added text descriptions of the corresponding stages in Figure 6, as per your suggestion.

---

### Author Response · Authors · 2023-11-21
**General Response**

We appreciate the reviewers' insightful feedback. It's encouraging that all the reviewers recognize and agree that this paper presents "highly impressive" "highly promising" results through "effective and convincing" techniques. In this paper, we point out the key role of hierarchical generation and bootstrapped diffusion prior which substantially improve the perceptual quality over the state of the art. As the Reviewer caAd highlighted, we strive for our method to represent a "great advancement" in the field.

We thank all the constructive review comments, which we found instrumental to further improving the exposition of the work. We have revised our manuscript accordingly to address and clarify the issues raised. To summarize, the paper has been improved in the following aspects:

- Provided additional qualitative results showcasing significant superiority over the state of the arts.

- Added comprehensive quantitative and qualitative ablation studies to delve deeper into the impact of the individual techniques.

- Explored the robustness of our method when applied to out-of-domain reference images, providing further evidence of the imaginative and combinational power of the method.

- Provided a detailed training time profile for stages.

- Improved clarity of the methodology section with additional implementation details (such as pseudo-algorithm) in the appendix.

- Conducted a user study to better reveal the perceptual quality that is hardly measured quantitatively.

We believe these improvements further elevate the quality and clarity of our paper, and we look forward to further review feedback.

---

### Meta-Review · Area_Chair_CZBe · 2023-12-07

**Metareview:**

This work proposes a hierarchical technique for 3D generation from text inputs using DreamBooth and SDS-like optimization schemes. Majority of the reviewers lean towards accepting the work. Reviewers appreciated the high-quality results. Several reviewers felt that the proposed work seems like a mix of existing works and expresses concerns over the technical novelty. Several of the clarity concerns and missing ablations are addressed in the author responses. The reviewers did raise some valuable concerns that should be addressed in the final camera-ready version of the paper, which include adding the relevant rebuttal discussions and revisions in the main paper. The authors are encouraged to make the necessary changes to the best of their ability. It is also important to release the code to advance further research in this area.

**Justification For Why Not Higher Score:**

Incremental novelty in technique w.r.t existing works.

**Justification For Why Not Lower Score:**

Multiple reviewers felt that the high-quality results outweigh the weaknesses of incremental novelty w.r.t. existing works.

---

### Decision · Program_Chairs · 2024-01-16

Accept (poster)